# Insights into a water-mediated catalytic triad architecture in CE20 carbohydrate esterases

Michelle Teune [1,6], Plínio S. Vieira [2,6], Thorben Döhler[1], Gottfried J. Palm [3], Theresa Dutschei[1], Daniel Bartosik [4], Leona Berndt[3], Gabriela F. Persinoti [2], Sandra Maaß [5], Dörte Becher [5], Thomas Schweder [4], Mário T. Murakami [2] ✉, Michael Lammers [3] ✉ & Uwe T. Bornscheuer [1] ✉

Carbohydrate esterases modify polysaccharides by removing different ester moieties thereby affecting their physicochemical properties and their accessibility by glycoside hydrolases. We determined the full-length structures of two members (Fl8CE20_II and PpCE20_II) from the carbohydrate esterase family 20 (CE20) by X-ray crystallography that feature an ancillary domain, inserted into the catalytic SGNH-hydrolase domain. Detailed structural analysis identifies a so far undescribed catalytic triad architecture which lacks the typical aspartate for polarization of the histidine but instead reveals a precisely coordinated water molecule mediating contact between the His and Asp. This coordinated water in the Ser-His-($H_2O$-Asp/Asn) motif, as further confirmed by mutational studies and by determination of kinetic constants, is crucial for catalytic activity. We therefore term this active site architecture a water-mediated catalytic triad.

In the search for alternative sources to fossil derivatives, plant biomass presents itself as a versatile and robust renewable material, contributing to the supply of high-value chemicals, pharmaceuticals and fuels[1–3]. For the conversion of organic matter, enzymatic methods are more sustainable and have the potential to replace traditional thermochemical methods, responsible for increasing production costs and harming downstream bioconversion steps[4]. In this context, CAZymes (carbohydrate-active enzymes) are essential to completely break down the complex net of carbohydrates that make up the biomass[5,6].

Of particular interest, carbohydrate-utilising bacteria possess an arsenal of CAZymes, usually organised in polysaccharide utilisation loci (PULs). These PULs are operons encoding relevant CAZymes for the depolymerisation of specific polysaccharides, accompanied by transmembrane transporters for oligo- and monosaccharides. These enzymes are critical in numerous biological processes such as biomass conversion, microbial metabolism, as well as cell wall remodelling and

can be found in various organisms such as plants, fungi and bacteria[7]. CAZymes include different classes of enzymes such as glycoside hydrolases (GHs) and polysaccharide lyases (PLs), which cleave glycosidic bonds, carbohydrate esterases (CE) that remove ester/amide-linked substitutions, and sulfatases that are especially important for marine carbohydrates that are likely to be highly sulfated[7]. Carbohydrate-binding modules (CBMs) assist in enzymatic breakdown by binding and recruiting specific carbohydrate substrates[8].

Carbohydrate esterases (CEs) hydrolyse ester or amide bonds in carbohydrate substrates, facilitating the degradation and modification of complex polysaccharides like chitin, pectin and hemicelluloses such as xylan[9]. In nature, CEs play vital roles in carbon cycling and plant-microbe interactions, while in industrial contexts, they are applied in fields such as biofuel production, animal feed, food processing, and pharmaceuticals[10,11]. The diversity of ester functionalities that need to be addressed for degradation, like *O*- and *N*-acetylations[12], ferulic acid[13]

[1]Department of Biotechnology & Enzyme Catalysis, Institute of Biochemistry, University Greifswald, Greifswald, Germany. [2]Brazilian Biorenewables National Laboratory (LNBR), Brazilian Center for Research in Energy and Materials (CNPEM), Campinas, São Paulo, Brazil. [3]Department of Synthetic and Structural Biochemistry, Institute of Biochemistry, University of Greifswald, Greifswald, Germany. [4]Department of Pharmaceutical Biotechnology, Institute of Pharmacy, University of Greifswald, Greifswald, Germany. [5]Department of Microbial Proteomics, Institute of Microbiology, University of Greifswald, Greifswald, Germany. [6]These authors contributed equally: Michelle Teune, Plínio S. Vieira. ✉e-mail: mario.murakami@lnbr.cnpem.br; michael.lammers@uni-greifswald.de; uwe.bornscheuer@uni-greifswald.de

or lignin[14], and their various positions within the polysaccharides, is reflected in the 20 CE families that are currently listed in the CAZy database based on sequence similarity, structure and substrate specificity[8].

Carbohydrate esterases use different catalytic strategies to act as de-*O*-acetylases and/or de-*N*-acetylases. Almost all CE families feature a conventional catalytic triad in their active sites. However, deviations from the canonical Ser-His-Asp/Glu configuration have been observed, for instance, in the CE4 family, which is harbouring a His-His-Asp triad and $Zn^{2+}$ or $Co^{2+}$ as a metal cofactor[15,16]. Very recently, the first dyad architecture observed for the CE4 family, which is able to coordinate the metal cofactor while lacking one His residue, was described[17]. The only CE family which is known for featuring a Ser-His dyad architecture for most members is the CE2 family. In these dyads, the His is supportably coordinated by the main chain carbonyl of a third residue, which is not crucial for catalytic activity[18].

Moreover, the enzymes differ in their oligomeric states, some acting as monomers while others are forming higher oligomers, such as CE7 enzymes acting as hexamers[19]. Members of families CE1, CE5, CE7 and CE15 share a α/β-hydrolase fold using a catalytic Ser-Glu/Asp-His catalytic triad. CE4, CE11 and CE14 enzymes are metalloenzymes, many of which use a catalytic $Zn^{2+}$-ion for catalysis and are structurally composed of a distorted $(α/β)_8$ fold, a two layer β-sandwich fold or a α/β-hydrolase fold, respectively[20]. Enzymes from families CE2, CE3, CE6, CE12, as well as CE20 share a SGNH hydrolase fold of the catalytic domain composed of repeated α/β/α-motifs resembling a Rossmann-fold[21] domain with a central six-stranded parallel β-sheet surrounded by α-helices on each side. This fold is characterised by four conserved residues: serine, glycine, asparagine, and histidine (the SGNH motif), arranged in a specific structural context that mediates the catalytic activity towards various substrates[22]. The SGNH-motif represents important catalytic residues as occurring in the primary sequence, i.e., Ser acting as nucleophile, Gly and Asn creating the oxyanion hole stabilising the tetrahedral reaction intermediate, and the His acting as base, increasing the nucleophilicity of the Ser by abstracting a proton. Notably, in CE20 enzymes, the Asn of the SGNH-motif is replaced by a Gln like it has been reported for CE6 enzymes before[20].

Among the carbohydrate esterases belonging to the SGNH hydrolase superfamily, the CE20 family was discovered back in 2021[23]. The crystal structure of the founding member of the CE20 family, XacXaeA from the plant bacterial pathogen *Xanthomonas citri*, suggested the SGNH hydrolase domain to be a discontinuous domain[24] by insertion of a domain annotated as X448 domain (here: ancillary domain). However, this domain was not resolved in the crystal structure[23]. XacXaeA is located in a xyloglucan utilisation locus and is capable to act as polysaccharide de-*O*-acetylase while being inactive as a de-*N*-acetylase. As the β-sandwich ancillary domain was not resolved in this structure, the exerted catalytic mechanism of these CE20 enzymes remained elusive[23].

Recently, three crystal structures of eukaryotic members of the CE20 family were described[25]. Notable, these eukaryotic CE20 family members do not contain an ancillary domain, resulting in significant differences in the active site architecture. For the mammalian sialic acid esterase, the catalytic His is bound and oriented by the main chain carbonyl of a Phe and not by an Asp/Asn, as observed in other enzymes using a Ser-His-Asp/Glu/Asn catalytic triad for catalysis[25]. A similar active site architecture was also reported for the SGNH superfamily extracellular lipase SrLip from *Streptomyces rimosus*, in which the main chain carbonyl of a Ser side chain orients the catalytic His rather than an Asp/Glu/Asn as observed in canonical Ser-containing catalytic triads[26].

In this study, we examined CE20 family members that feature an ancillary domain inserted into the SGNH hydrolase domain. Sequence similarity network (SSN) and gene neighbourhood analyses indicated a potential clustering of these enzymes in conjunction to their predicted

substrates, which was investigated in silico as experimental validation remains insufficient so far, due to a lack of suitable acetylated substrates. We were able to crystallise two of these enzymes, Fl8CE20_II from *Flavimarina* sp. Hel_I_48 and PpCE20_II from *Pedobacter psychrotolerans*, representing the full-length structures of the ancillary domain containing CE20 enzymes. These structures now provide important insights into the active site architecture of these enzymes that differ from the conventional Ser-His-Asp/Glu/Gln catalytic triad and they contain a precisely coordinated water molecule in the active site, which we name water-mediated catalytic triad. Our detailed mechanistic analyses of the active site architecture of these ancillary domain-containing CE20 enzymes provide fundamental knowledge about carbohydrate esterase function, its potential role in mediating substrate specificity and on convergent enzyme evolution.

## Results
### Sequence similarity analyses and substrate prediction for enzymes of the CE20 family
Enzymes belonging to the CE20 family display a distinctive domain architecture, comprising an N-terminal and a C-terminal β-sandwich domain, each composed of an antiparallel seven-stranded β-sheet flanking a central catalytic SGNH hydrolase domain. Certain CE20 enzymes comprise an additional β-sandwich ancillary domain inserted into the discontinuous SGNH hydrolase domain, thereby splitting it into two structural units[23]. The ancillary domain is connected by two linker sequences (Fig. 1a). However, in the tertiary structure, both sections of the SGNH domain form a monolithic structure. The CE20 β-sandwich ancillary domain is directly adjacent to the SGNH-hydrolase fold domain and is involved in active site formation.

To further investigate the role of this ancillary domain, we performed sequence-similarity network (SSN) analyses by clustering protein sequences from the CE20 family containing an ancillary domain. By using the sequence of the founding member XacXaeA, several sequences, all from bacterial origin, could be retrieved from the UniProt database[27]. The resulting SSN (Fig. 1b) revealed that these sequences could be separated into six clusters. Sequences of the main cluster (cluster I) show a diffuse distribution, while the remaining five clusters (clusters II to VI) are precisely defined. The clusters I, II, IV and V contain sequences of the phylum Bacteroidota, while clusters III and VI contain sequences from the phylum Proteobacteria (Pseudomonadota).

Although sequences among selected members of all the clusters found in the SSN have an identity of around 38% (Supplementary Table 1), it is striking that, when analysed separately, the catalytic domain has a higher identity (average 43%), while the internal ancillary domain is more variable (average 29% identity). Only when comparing two sequences from the same cluster (Fl8CE20_II and PpCE20_II), the similarity of the ancillary domain is comparably high as the overall similarity (42%). From this, the hypothesis emerged that this ancillary domain might play a crucial role in substrate recognition.

For further classification of the clusters, the genetic neighbourhood of selected representing enzymes from each cluster was examined to infer potential substrate specificities, leading to the cluster classification based on putative substrate specificities (Fig. 1b). For cluster I, a CE20 from *Bacteroides nordii* (BnCE20_I) was selected. Most co-occurring genes are related to pectin hydrolysis (GH28, GH36, GH106, CE19), followed by external membrane sugar transporter genes (*susC/D*). For cluster II, the CE20 from *Pedobacter psychrotolerans* (PpCE20_II) is encoded next to xylan processing enzymes (GH10, GH43_1, GH67, GH115, CE15) and the Fl8CE20_II from *Flavimarina* sp. that is located in a PUL which is already elucidated on targeting glucuronoxylans (GH27, GH43_10, GH43_29, GH95, GH115, CE1, CE6, where Fl8CE20_II specifically shows activity towards acetylated xylans (Fig. 1b))[28]. XacXaeA, the founding member of the CE20 family[23], is present in Cluster III and was already characterised and found in a

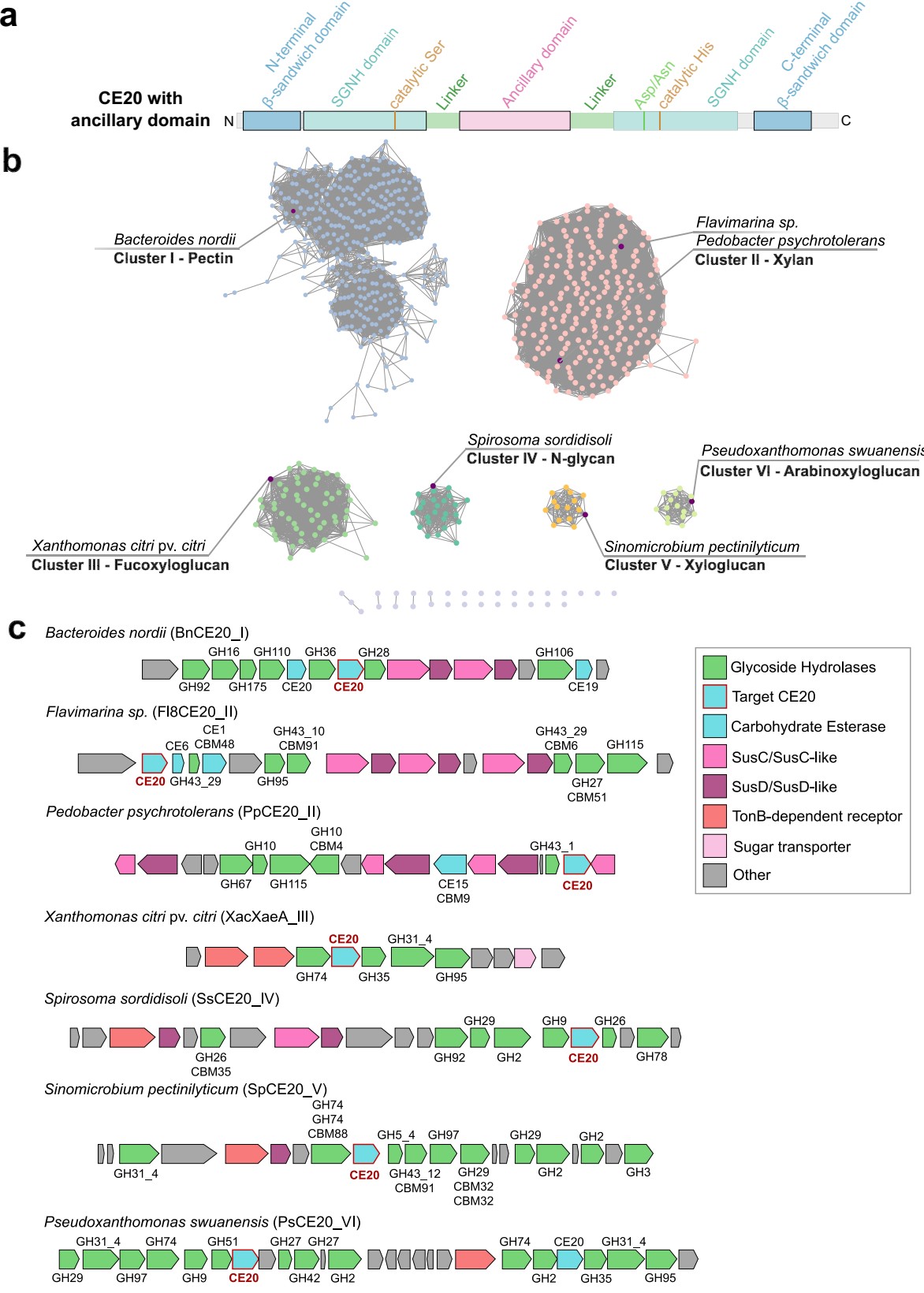

fucogalactoxyloglucan gene operon from *Xanthomonas citri* pv. *citri* (GH31_4, GH31, GH35, GH74, GH95), with activity on oligosaccharides from this hemicellulose (XacXaeA_III). In cluster IV, the esterase gene for *Spirosoma sordidisoli* (SsCE20_IV) is accompanied by genes that encode for enzymes related to *N*-glycan degradation (GH2, GH9, GH26, GH29, GH78). *Sinomicrobium pectinilyticum* (SpCE20_V) in

cluster V shows a co-occurrence of genes that could address different types of xyloglucans (GH2, GH3, GH5_4, GH29, GH31_4, GH43_12, GH74). Finally, cluster VI contains the CE20 target sequence from *Pseudoxanthomonas wuyuanensis* (PwCE20_VI), where neighbourhood genes seem to encode for proteins that might act on arabinox-yloglucan (GH2, GH9, GH27, GH29, GH31_4, GH51, GH74). In terms of

**Fig. 1 | Sequence similarity network (SSN) of CE20 enzymes and genomic neighbourhood analyses of selected sequences of each cluster. a** Domain architecture of CE20 enzymes featuring the ancillary domain. The primary sequence of the SGNH hydrolase domain is interrupted by the ancillary domain. Both domains are connected by a linker motif. The SGNH hydrolase domain is flanked by an N- and a C-terminal β-sandwich domain. **b** SSN from the CE20 family sequences that possess the ancillary domain. Each cluster is represented in different colours, and the selected target is marked in dark purple. The original source microorganism is indicated, as well as the number of the cluster and the substrate (putative: I, IV, V and VI, empirically proven II and III) associated with it. Each dot represents an individual CE20 sequence. **c** Co-occurrence analysis of the genomic context, showing which GH families are present in the PUL (light green). Genes encoding for esterases (light blue) aside CE20 (light blue, highlighted by red edge) are also indicated. Transporters are assigned (pink: SusC/SusC-like, purple: SusD/SusD-like, red: TonB-dependent receptors, light pink: internal sugar transporters). Other genes unrelated to carbohydrate hydrolysis are depicted in grey.

biochemical and kinetic behaviour, all enzymes showed similar properties (Supplementary Fig. 1 and Supplementary Table 2).

To investigate how this variance in the ancillary domain might modulate substrate binding, sequence alignments of the seven representatives were performed (Supplementary Fig. 2). These alignments initially revealed that a loop region, lining the active site in the SGNH hydrolase domain (based on AlphaFold3 models), shows high diversity across the seven sequences of the representative enzymes (Supplementary Fig. 3). While the potential xylan targeting cluster II enzymes revealed five highly conserved residues (Fl8CE20_II: Thr/Ser331, Y332, Y334, P335, R337) in this loop, the fucoxyloglucan cluster III only showed a strictly conserved aromatic side chain, i.e., Trp/Tyr (XacXaeCE20_III: Trp329), a conserved Asn (XacXaeCE20_III: Asn330) and a moderately conserved Pro (XacXaeCE20_III: Pro332) structurally located in direct vicinity to the catalytic SGNH hydrolase domain. Xyloglucan cluster V enzymes also show a conserved Trp (SpCE20_V: Trp337), while the Asn (SpCE20_V: Asn40) in this loop is less conserved and replaced by an Asp/Glu/Ala in some members. Instead, in this cluster, a Ser/Thr (SpCE20_V: Ser39) shows a high conservation score. Enzymes of the arabinoxyloglucan targeting cluster VI show four highly conserved residues forming a Cys-Gly-Trp/Tyr-Asp/Glu motif (PsCE20_II: Cys328; Gly329; Trp330; Glu331). The aromatic and the acidic residue can vary, being exchanged with residues of similar physicochemical properties in this motif. While all enzymes present in xyloglucan-associated clusters III, V and VI show a conserved Trp at one position, enzymes of the xylan targeting cluster II exclusively contain one to two conserved Tyr residues within this loop. In general, most of the conserved residues of the ancillary domain are not directly contacting residues within the SGNH hydrolase domain, indicating that they are not primarily needed to mediate interaction and orientation of the two domains. Instead, these residues, pointing into the active site cleft between the ancillary domain and SGNH hydrolase domain, might be involved in binding and coordinating poly- and oligosaccharide substrates.

Notably, all enzymes exhibited some degree of activity on semi-synthetic acetylated xylan which was the only acetylated polysaccharide available (Supplementary Fig. 4).

## Fl8CE20_II homologues show further association with xylan degradation

Fl8CE20_II was previously proven to solely act as a de-O-acetylase on acetylated xylans. No other xylanolytic esterase activity, like feruloyl esterase or glucuronylesterase activity, was observed (Supplementary Fig. 5). Previous studies of CE20 enzymes also indicate that enzymes of this CE family only act as de-O-acetylases acetylesterases[8,23,25]. Biochemical characterisation of the enzyme shows a preference for slightly basic pH-values, a high tolerance towards $Ca^{2+}$, $Cu^{2+}$, $K^+$, $Mg^{2+}$ ions, and remaining activity of 40% at 2.5 M NaCl (Supplementary Fig. 6). Furthermore, previous studies revealed that the expression of Fl8CE20_II is upregulated in *Flavimarina* sp. Hel_I_48 upon growth on different xylans as a carbon source, while for pectin, which is as well acetylated, no upregulation was detected (Supplementary Fig. 7)[28]. Thereby, the preference of Fl8CE20_II towards xylans is sufficiently understood and was selected as a basis for the identification of homologous proteins representing

the xylan targeting cluster II. Phylogenetic analysis of these sequences indicated a predominance of bacterial species belonging to the phylum *Bacteroidota* with 359 out of 502 identified sequences (Supplementary Fig. 8). It is worth noting that orthologues to Fl8CE20_II were also found distributed across 18 bacterial phyla (Supplementary Fig. 9). Interestingly, some of these enzymes feature additional domains such as putative CE4 domains, CE15 domains and in one case, a GH10 domain (Supplementary Fig. 8). Since CE15 and GH10 enzymes are known to be xylanolytic CAZymes[8] this matches the association of cluster II with xylan processing. This can be further supported by analysing the genomic context of these homologues (Supplementary Fig. 10), which co-localises with GH10, GH67, GH43 or GH3 encoding genes and by the fact that the most predicted substrate for these sequences are xylans (Supplementary Fig. 10a). Moreover, the carbohydrate binding modules (CBMs) CBM91, CBM4, CBM6, and CEs CE1, CE15 encoded in the vicinity to cluster II genes are mostly associated with xylan degradation (Supplementary Fig. 10b). These analyses again suggest that the cluster II enzymes are primarily associated with xylan breakdown.

Future investigation of the diversity in CE20 enzymes featuring the ancillary domain may provide a deeper understanding of the role of this domain, which appears to vary across the SSN clusters.

## The β-sandwich ancillary domain is involved in active-site cleft formation

The selected protein sequences from the SSN were studied regarding their structural properties. To this end, the three-dimensional structures of two proteins were obtained by crystallography: Fl8CE20_II (PDB 9H4U, 1.54 Å) and PpCE20_II (PDB 9EGA, 1.35 Å) (Supplementary Table 3 and Supplementary Fig. 11). In contrast to the previously reported structure of *Xanthomonas citri* XacXaeA (PDB 7KMM), for which the ancillary domain was not resolved, both structures cover all domains of the enzymes.

The structures of PpCE20_II and Fl8CE20_II have an identical domain organisation and a highly similar overall fold (RMSD 1.151 Å) with a scorpion-like shape due to the geometric orientation of the internal β-sandwich ancillary domain in relation to the central catalytic domain connected by two linkers resembling a tail (Fig. 2a, b). Due to their high similarity, only the structure of Fl8CE20_II is shown in Fig. 2. The structure of PpCE20_II can be found in the supplementary information (Supplementary Fig. 11).

As mentioned, those CE20 family members like Fl8CE20_II share a modular catalytic domain divided in two halves by insertion of a β-sandwich ancillary domain. In the tertiary structure, the two halves of the catalytic domain form a central single monolithic SGNH-hydrolase domain. The catalytic SGNH hydrolase domain is flanked by an N-terminal and a C-terminal β-sandwich domain each composed of a seven-stranded antiparallel β-sheet on each side (Fig. 2a, b). The ancillary domain is inserted into the SGNH hydrolase domain by an α-helical linker resulting to position the ancillary domain over the catalytic domain forming the substrate binding region thereby partially closing it (Fig. 2b and Supplementary Figs. 11 and 12). The ancillary domain of Fl8CE20_II has a β-sandwich fold, composed by eight antiparallel β-strands, reminiscent from β-galactosidases of the GH2 family (PFAM PF02837 or PF13364). Analysis of the surface electrostatics

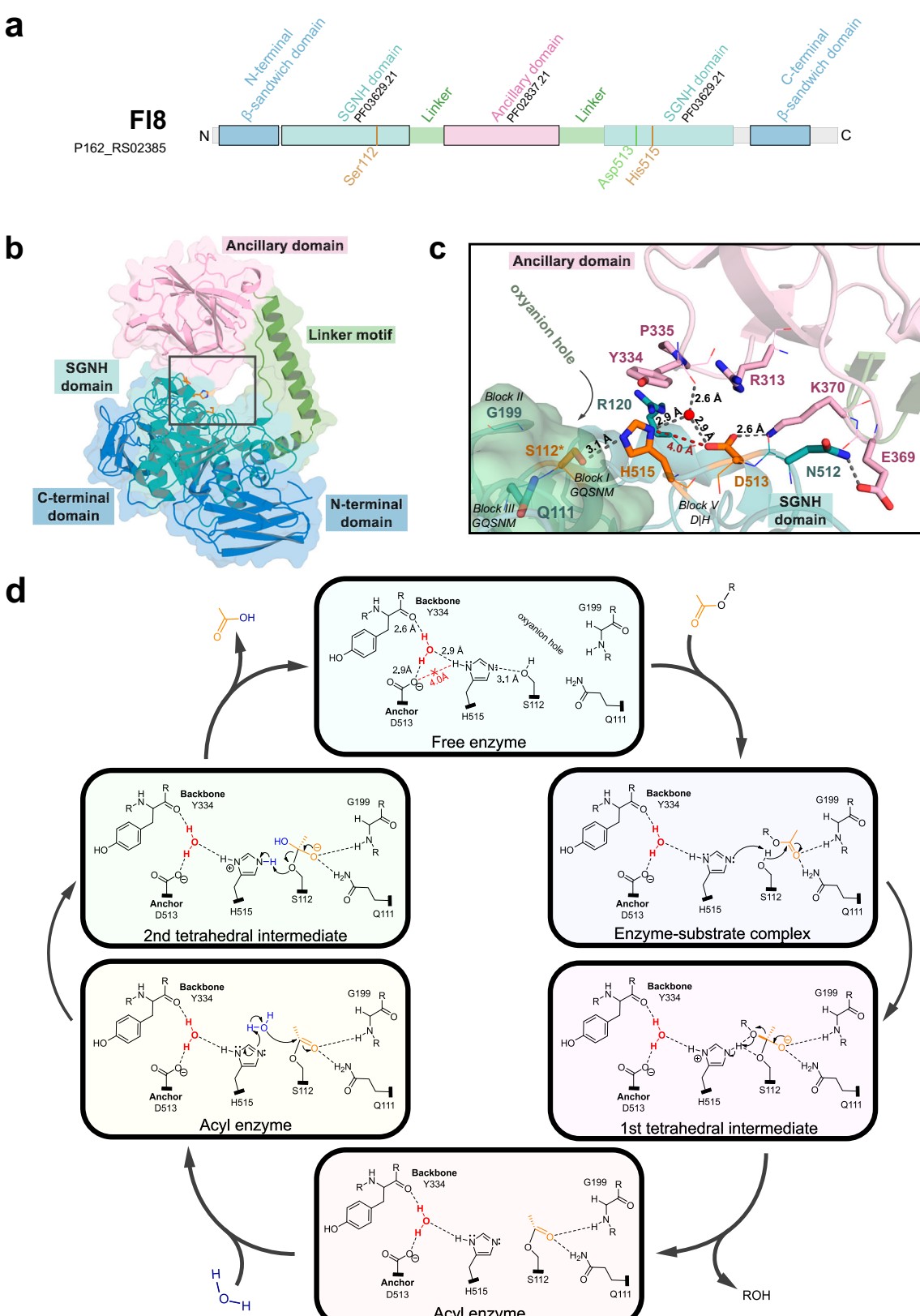

shows the ancillary domain is highly positively-charged, containing several Lys and Arg side chains (Fl8CE20_II: Arg313; Lys370; PpCE20_II: Arg314; Lys371) facing the active site of the SGNH hydrolase domain (Fig. 2a and Supplementary Fig. 11). This enables the establishment of salt bridges and hydrogen bonds with residues in the SGNH hydrolase domain. (Fig. 2b). In the active site and proposed substrate binding cleft, formed by the SGNH hydrolase domain and the ancillary domain, the expected catalytic residues (Ser-His-Asp) lie partially exposed to the solvent (Fl8CE20_II: Ser112, His515, Asp513; PpCE20_II: Ser114, His520, Asp518. The presence of highly conserved aromatic side chains in the aforementioned loop region formed at the interface between the ancillary domain and SGNH hydrolase domain furthermore suggest

**Fig. 2 | Structure and mechanism of the Fl8CE20_II. a** Domain architecture including Pfam annotations and (**b**) crystal structure of Fl8CE20_II. Cartoon and surface representation of the catalytic domain (shades of blue) and the ancillary domain (light pink). Both domains are connected by a linker motif (green) that forms a hoop. The water-mediated catalytic triad, including the anchor residue, are represented as orange sticks. **c** Details of the active site and anchoring of the ancillary domain over the catalytic domain, with relevant residues shown as sticks. The Ser is cacodylate-modified (omitted for clarity) due to crystallisation conditions. Naturally occurring modification of the Ser was precluded via MS analysis of the purified Fl8CE20_II (data not shown). Black dashed lines represent hydrogen bonds. The red dashed line and the distance highlighted in red indicate the

prohibitive distance for the right protonation of the catalytic His by the Asp. The numbers above the dashed lines show the distances between hydrogen bond donor and acceptor atoms. The coordinated water is represented as a cyan sphere. **d** Proposed catalytic mechanism for CE20 enzymes harbouring the ancillary domain. The residue numbering corresponds to the Fl8CE20_II primary sequence. The orange compound is the acetyl group attached (*R*)-configured to the oligo-saccharide. The coordinated water is represented in red, and catalytic water is highlighted in blue. Electron density maps of the active sites of Fl8CE20_II and PpCE20_II can be found in the Supplementary Fig. 16. Omit maps of the active sites of Fl8CE20_II and PpCE20_II can be found in the Supplementary Fig. 17.

that these might be involved in substrate binding, as it is also known for members of the CE2 family[18,29].

## CE20 members operate via a water-mediated catalytic triad

In classical SGNH hydrolase enzymes, the aspartate of the catalytic triad (corresponding to D513 in Fl8CE20_II) acts as a proton acceptor polarising and orienting the catalytic histidine (H515 in Fl8CE20_II), in the D<u>x</u>H motif (Block V), which in turn deprotonates the catalytic serine, (S112 in Fl8CE20_II) (GQ<u>S</u>NM motif in Block I) thereby increasing its nucleophilicity. In analogy to other catalytic triad architectures, including α/β-hydrolases and other proteases and esterases, all three residues are expected to be in the hydrogen bond distance of 2-3 Å resulting in an effective activation of the Ser nucleophile.

In both crystal structures of Fl8CE20_II and PpCE20_II, representing the only structures of CE20 enzymes featuring the inserted ancillary domain solved so far, the Asp (Fl8CE20_II: Asp513; PpCE20_II: Asp518) carboxylate has a distance to the catalytic His Nδ of 4.0 Å which is too far away for a direct interaction (Fig. 2c and Supplementary Fig. 11). A direct interaction between the His base and Asp/Asn to occur during catalysis can be excluded as a distance of < 3 Å cannot be achieved rotating the side chains, i.e., χ1 and/or χ2 of the His base and/or the Asp/Asn (Supplementary Table 4). To this end, a conformational alteration of the main chain would be needed to allow direct contact. However, this is unlikely to occur as the *B*-factors for the Cα-atoms of the catalytic residues Ser-His-Asp are all between 17.7 to 29.7 Å² for both chains, confirming these are structurally well defined with a low degree of flexibility (Supplementary Table 4).

This unusual distance fed our hypothesis that this Asp is not directly involved in positioning or polarisation of the catalytic His (Fig. 2c and Supplementary Fig. 10e). Instead, the Asp side chain contributes to precisely coordinating a water molecule (Fl8CE20_II: 2.9 Å), which is in direct hydrogen bond distance to the catalytic His base (Fl8CE20_II: 2.9 Å). The tetrahedral coordination geometry of this water molecule is completed by interaction with a main chain carbonyl oxygen of Tyr334 in the ancillary domain and by another water molecule. In some of the selected CE20 enzymes from the SSN, this aspartate can be naturally replaced by an asparagine as well (Supplementary Fig. 13). The asparagine side chain can also establish a hydrogen bond to coordinate the water molecule. We propose that the precise coordination of the water molecule in this, so far undescribed, active site architecture is crucial for catalytic activity, as the water molecule is involved in the coordination of the histidine. Since this catalytic triad architecture involves a water molecule precisely coordinated next to the histidine with several hydrogen bonds involved, we termed this a water-mediated catalytic triad, i.e., Ser-His-(H₂O-Asp/Asn) (Fig. 2c).

This architecture is different to the CE20 enzymes not featuring the ancillary domain. In comparison, structures of eukaryotic sialic acid esterase, assigned to family CE20, where the ancillary domain is not inserted into the catalytic domain, were recently solved and reported to have a Ser-His catalytic dyad[25]. In those proteins, a loop bends close to the active site, where the backbone of an aromatic

residues interacts directly with the catalytic histidine without requiring a water molecule in between the residues (Supplementary Fig. 14). As in the dyad serine proteases or in many cysteine proteases, the third residue has been reported to be not essential for an efficient catalysis and can be replaced as the side chain is often not involved in hydrogen bond formation[26,30]. However, this definition is inconsistent in the literature. For some dyads, like papain, mutating the third residue leads to a strong reduction in activity, yet the structure is still commonly described as a dyad (Supplementary Fig. 15)[31].

Notably, although a catalytic triad including a water molecule in the active site can be observed also for proteases and other enzymes such as chalcone synthase, the structural context and the active site geometry is different in CE20 enzymes and was not described so far[32–35]. Screening the protein data bank for similar structural motifs using real-time structural motif search (RMSD cutoff: 2 Å) only resulted in one sequence, the founding member of the CE20 family, XacXaeA[23], indicating that this exact active site architecture is exclusive to this CE family.

Accordingly, we propose the following catalytic mechanism for the CE20 enzymes with ancillary domain using this water-mediated catalytic triad: the His base (Fl8CE20_II: His515; PpCE20_II: His520) is coordinated by the active site water molecule, which is in turn coordinated by the Asp/Asn (Fl8CE20_II: Asp513; PpCE20_II: Asp518; called anchor residue as it forms interactions to the ancillary domain) and the main chain carbonyl group of the ancillary domain (Fl8CE20_II: Tyr334; PpCE20_II: Tyr335). His activates the Ser (Fl8CE20_II: Ser112; PpCE20_II: Ser114) in turn, exerting nucleophilic attack on the substrate. Subsequently, a first tetrahedral intermediate is formed with a negative charge at the oxygen which is stabilised by the oxyanion hole formed by the main chain amide of a Gly side chain and the amide side chain of a Gln (Fl8CE20_II: Gly199 of block I; Gln in GQ<u>S</u>NM motif in Block III) (Fig. 2c).

After resolving the tetrahedral intermediate catalysed by the His – now acting as acid – a covalent acyl-enzyme intermediate is formed while the deacetylated carbohydrate is released. The covalent acyl-enzyme intermediate is subsequently resolved this time using a second water molecule as nucleophile, activated by the His base. A second tetrahedral intermediate is formed, which is resolved in analogy to the first tetrahedral intermediate, finally resulting in the released acyl-group and the regenerated catalytic residues (Fig. 2d). To further validate our hypothesis derived from the structural analyses, we next performed mutational studies.

To further support the hypothesis of a catalytic relevant function of the coordinated water molecule experimentally, mutational investigations of Fl8CE20_II were performed (Fig. 3). The structural studies showed that the water molecule adopts a tetrahedral coordination geometry by interacting with the active site Asp513, with Tyr334 of the ancillary domain and with a second water molecule (not shown for better visibility). Moreover, besides coordinating the water molecule, Asp513 in the SGNH hydrolase domain forms a direct salt bridge to Lys370 of the ancillary domain. Thus, Asp513 has a dual function, i.e., a catalytic role in coordinating and positioning the catalytic water

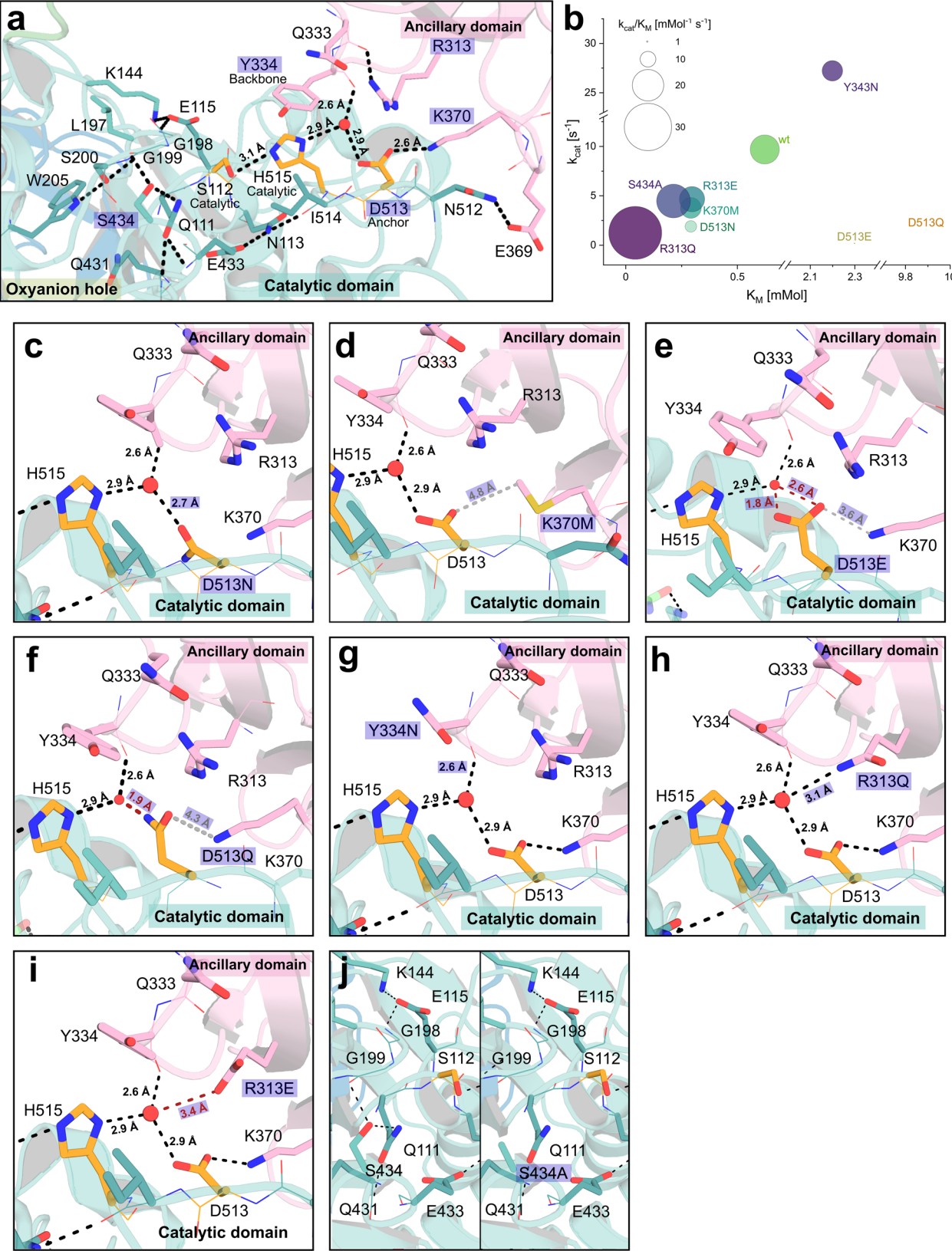

molecule and a structural role in binding and orienting the ancillary domain with direct consequences on substrate binding. Several residues that interact with the coordinated water (D513, Y334) and/or form hydrogen bonds between the ancillary and catalytic domain (K370) were chosen for mutational studies (Fig. 3a) and investigated regarding their kinetic parameters (Fig. 3b and Supplementary Table 5).

Mutation of the anchor Asp in Fl8CE20_II to Asn (D513N, Fig. 3c) mimics the natural variation of CE20 enzymes at this position. Asp and Asn are both capable of forming the hydrogen bond network to coordinate and position the water molecule, however, the impact on polarisation of the water molecule might be slightly different (Fig. 3b, c and Supplementary Table 5). Compared to the Fl8CE20_II wildtype, the

**Fig. 3 | Mutational analysis of relevant residues of Fl8CE20_II. a** Closeup of the active site region, showcasing mutated residues (highlighted in a purple box) and relevant residues that directly interact with them through hydrogen bonds. The coordinated water is represented as a red sphere. **b** Kinetic parameters of selected residues, represented as bubble chart, where the $X$- and $Y$-axis represent $K_M$ and $k_{cat}$, respectively, and the bubbles represent the catalytic efficiency in $k_{cat}/K_M$. These experiments were conducted as three independent technical replicates ($n = 3$). Mean values ± SD are shown in Supplementary Table S5. Selected mutated regions,

as shown in (**a**), displaying how changing the residue alters important hydrogen bonds: D513N (**c**), K370M (**d**), D513E (**e**), D513Q (**f**), Y334N (**g**), R313Q (**h**), R313E (**i**) and S434A (**j**). Mutated residues and changes in H-bond distances are highlighted in a purple box. The tertiary structures are represented as cartoons, side chains as coloured sticks (pink: ancillary domain, light teal: catalytic domain, orange: catalytic residues). Dashed lines indicate possible polar interactions (black), potential sterically hindrance (red) and potential loss of interaction due to increased distance (grey).

mutant D513N shows a slightly decreased (~ 2.5-fold) catalytic efficiency $k_{cat}/K_M$. This difference is mostly due to the fivefold reduced turnover number $k_{cat}$ rather than the $K_M$ value, suggesting the mutation has a slight impact on catalysis with little effect on substrate binding. This slightly reduced catalytic turnover ($k_{cat}$) observed for the mutant Fl8CE20_II D513N might be due to the loss of the salt bridge with Lys370 from the ancillary domain interfering with positioning of Asp513. Supporting this, the mutation of Lys370 to Met (K370M, Fig. 3d) also resulted in a slight reduction of $k_{cat}$ while showing the same $K_M$ as the mutant D513N. This suggests that the D513N mutation slightly affects catalysis rather than substrate binding, maybe due to a different capacity to polarise the water molecule.

As expected, the mutation Asp513 to Ala in Fl8CE20_II (D513A) resulted in complete inactivation of the enzyme showing the importance of this residue for catalysis (Supplementary Table 5). As Asp513 is on the one hand a constituent of the catalytic active site and on the other hand it directly interacts with the ancillary domain, mutation of Asp513 to Ala might interfere with $K_M$ as an indicator for substrate binding and $k_{cat}$ as an indicator of catalytic activity. To this end, this mutation does not allow us to deduce the importance of the catalytic water molecule. Thus, we constructed another mutant, i.e., Fl8CE20_II D513E (Fig. 3e), with a Glu conserving the negative charge and the salt bridge to Lys370 but being sterically more demanding compared to Asp with consequences also on positioning the active site water molecule. We observed this mutant does result in a mild impact on substrate binding, lowering $K_M$ by 3.7-fold (WT: 0.62 mM; D513E: 2.30 mM) while strongly impairing $k_{cat}$ (WT: 9.70 s$^{-1}$; D513E: 0.43 s$^{-1}$). Mutation of the Asp to Gln (D513Q, Fig. 3f) almost loses activity indicated by a 16-fold increase in $K_M$ (WT: 0.62 mM; D513E: 9.89 mM) and decreased $k_{cat}$ (WT: 9.70 s$^{-1}$; D513Q: 0.18 s$^{-1}$). This supports that Asp513, unlike in typical dyad architectures, is indeed involved in catalysis next to its role on substrate binding and supports the validity of the proposed water-mediated Ser-His-($H_2O$-Asp/Asn) catalytic triad architecture.

Interestingly, the mutation of Tyr334 to Asn in the ancillary domain resulted in a decrease in $k_{cat}$ and an increase in $K_M$ (Fig. 3b, g and Supplementary Table 5). This Tyr is highly conserved among cluster II enzymes supporting its important role in catalysis and xylan substrate binding (Supplementary Fig. 3). Of all mutants analysed, apart from D513A, the mutant R313Q had the strongest negative impact on the $k_{cat}$ (WT: 9.7 s$^{-1}$; R313Q: 1.3 s$^{-1}$), which might be caused by the potential interference in the hydrogen bond network of the coordinated water (Fig. 3b, h and Supplementary Table 5). R313 might be able to form hydrogen bonds to the coordinated water molecule, altering its ability for a potential polarisation of the catalytic His, causing a decrease in $k_{cat}$. However, this is compensated by an almost 15-fold decreased $K_M$, i.e., improved substrate binding (WT: 0.62 mM; R313Q: 0.04 mM). The mutant R313E did not interfere with the hydrogen bond network (Fig. 3i), resulting in only slightly altered $k_{cat}$ (WT: 9.7 s$^{-1}$; R313E: 4.67 s$^{-1}$) and $K_M$ (WT: 0.62 mM; R313E: 0.30 mM). Mutation of S434, located in the oxyanion hole next to Q111, to Ala (Fig. 3j) resulted again in a small decrease of $k_{cat}$ (WT: 9.7 s$^{-1}$; S434A: 4.48 s$^{-1}$) and $K_M$ (WT: 0.62 mM; S434A: 0.21 mM).

For most mutants located in the interaction interface between the ancillary domain and the SGNH hydrolase domain, we observed the general trend that these mutants show a comparable catalytic

efficiency $k_{cat}/K_M$ compared to wild-type Fl8CE20_II. An impaired catalytic activity, $k_{cat}$, is compensated by an improved substrate binding affinity, $K_M$ and vice versa (Fig. 3b).

Overall, these detailed mutational studies suggest a crucial role of both the orientation of the ancillary domain in the right position to the catalytic domain and the precise coordination of the water molecule for catalytic activity. As the ancillary domain is, as well as the D513, involved in the coordination of the water molecule, it is challenging to differentiate between both functionalities, as every mutation affecting the water molecule additionally also interfers with the orientation of the ancillary domain. Nevertheless, the water-mediated catalytic triad clearly distinguishes from Ser-His dyads using a main chain NH at the third position or conventional triads, that feature a direct interaction between the acid and the His (Supplementary Fig. 15)[25,26,36,37].

## Discussion

We describe the structure-function analyses of two members from the CE20 family, i.e., Fl8CE20_II from *Flavimarina* sp. and PpCE20_II from *Pedobacter psychrotolerans*, featuring an ancillary domain inserted into the SGNH hydrolase domain. Based on our SSN analyses both CE20 enzymes are categorised into cluster II associated with xylan breakdown and metabolism. They represent the first crystal structures from the CE20 family featuring the ancillary domain which could be solved in their full-length forms. An earlier reported structure of the family XacXaeA in 2021 did not encompass the β-sandwich ancillary domain due to proteolysis while known structures of eukaryotic sialic acid esterases do not feature this ancillary domain[23,32]. These studies agree that enzymes of the CE20 family share a catalytic core domain consisting of a central catalytic SGNH hydrolase domain flanked by β-sandwich domains at both sides. In Fl8CE20_II and PpCE20_II, the SGNH hydrolase domain is discontinuously[24] organised by insertion of the ancillary domain, which is only present in some, to current knowledge, mainly bacterial CE20 members.

Sequence based investigations of the ancillary domains present in CE20 enzymes of different clusters could indicate their relevance for substrate recognition and binding. Future investigations of the preferences of each cluster and the involvement of the conserved loop region residues will be essential to potentially offer insights into the evolutionary advantage deriving from the insertion of this ancillary domain.

The core observation of this work is an unconventional active site architecture observed in both crystal structures. In contrast to a conventional catalytic triad, the distance between the catalytic Asp/Asn and the catalytic His is too far to establish a direct interaction. Instead, the conserved Asp/Asn in this class of CE20 enzymes uses a precisely coordinated water molecule to indirectly contact the catalytic His. The tetrahedral coordination geometry of the water molecule is completed by binding to the main chain carbonyl of a conserved Tyr in the ancillary domain and by another water molecule. The fact that the Asp can be replaced by Asn in some enzymes suggests the water molecule mostly being important to position the His rather to polarise it. Our mutational data confirms the correct placement of the water molecule being indeed important for enzymatic activity.

Notably, a water-mediated catalytic triad active site configuration is present in five Ser protease families as classified in MEROPS (S54,

S59, S68, S78 and S80)[38]. For rhomboid Ser proteases of the MEROPS family S54, a similar active site architecture with a water molecule bond by an Asp and directly interacting with the His base was reported, but not further investigated[34]. Also 3C-like proteinase of SARS-CoV-2 shows a similar architecture, featuring a water molecule between an Asp and the catalytic His, but this was considered a dyad without further investigation[32,33]. However, this active site architecture is realised in a substantial different geometry and structural context compared to CE20 enzymes with an SGNH hydrolase domain. As the terms triad and dyad are used inconsistently in the literature, a clear distinction of the water-mediated triad from known active sites is necessary.

Inspection of enzymes reported to use a catalytic dyad rather than a triad for catalysis, i.e., lacking an Asp/Glu/Asn directly contacting the active site His, revealed in these enzymes the His being positioned and/ or polarised by the main chain carbonyl group, i.e., in glycerinaldehyde-3-phosphate dehydrogenase[39–41] or *Streptomyces scabies* esterase[37], the main chain NH groups, i.e., in aspartate semialdehyde dehydrogenase[42] or cysteine desulfurase[43] or by a water molecule as observed in chalcone synthase[35], rhomboid protease[34] or SARS-CoV-2 3C-like proteinase[32,33]. Even in well-studied Cys proteases such as papain[31,44,45], cathepsin B[46,47] or legumain[48], for which a catalytic dyad is reported, the His of the dyad is in direct interaction distance to an Asn. For papain, this interaction was reported to be important for full catalytic activity[31,45].

In conclusion, we could clearly show that the active site Asp/Asn side chain of Fl8CE20_II is in fact, crucial for catalytic activity, while it is, in contrast to a conventional triad, not directly interacting with the His. Accordingly, in light of the significant differences between the so far described dyads and triads and especially in combination with our structural and mechanistic analyses supported by in-depth mutational studies, we suggest to term this so far undiscovered mechanism a water-mediated catalytic triad realised in CE20 enzymes. The fact that this active site architecture was realised in different enzymes with different structural contexts[33–35,48] suggests this water-mediated catalytic triad being independently developed, giving rise to an example for convergent enzyme evolution.

## Methods

### Generation of Fl8CE20_II single point mutations

The gene of Fl8CE20_II from *Flavimarina* sp. Hel_I_48 (P162_RS02385) was cloned into a pET28a(+) vector generating the pET28a(+) _Fl8 as previously described[28]. Primers for site-directed mutagenesis were designed using the NEBaseChanger® tool. PCRs were performed in 25 μL scale using Q5 High Fidelity DNA Polymerase (NEB®) as suggested by the manufacturer. The PCR product was used for the kinase, ligase, and *Dpn*I reaction (KLD Enzyme Mix, NEB®) for 1 h at room temperature (RT). 2 μL were used for the transformation of *E. coli* TOP10 cells. Plasmid isolation was performed, and the desired mutations were confirmed via Sanger sequencing. The plasmids were transformed in *E. coli* BL21 (DE3) for protein expression.

### Protein production and purification

Fl8CE20_II and mutants were recombinantly produced in LB-broth (Miller, Sigma Aldrich®) and purified via His-Tag as described previously[28]. The pET28a(+) plasmids of all enzymes (1 ng) were transformed into *E. coli* BL21 (DE3). A single colony was inoculated with LB-media and kanamycin (50 μg mL$^{-1}$) overnight. The expression was performed using LB-broth (Miller, Sigma Aldrich®) with kanamycin (50 μg mL$^{-1}$), baffled flasks and a culture to flask volume ratio of 1:5. The OD$_{600nm}$ was initially set to 0.1, and the culture was incubated at 37 °C, 140 rpm until an OD$_{600nm}$ of 0.6 to 0.8 was reached. The culture was cooled to 20 °C, and 0.5 mM IPTG was added. After 16–20 h the cells were harvested (450 x *g*, 30 min, 4 °C). The lysis was performed using ultrasonication on ice (2 × 3 min, 50 % power, 50 % cycle time) followed by centrifugation (10,000 x *g*, 10 min, 4 °C). The supernatant

was applied to ROTI®Garose-His/Ni Beads (Roth) with a column volume (CV) of 4 mL equilibrated with lysis buffer (50 mM Tris-HCl pH 8.0, 300 mM NaCl). The column was washed with 20 CV 20 mM imidazole, 50 mM Tris-HCl pH 8.0, 300 mM NaCl and the enzyme was eluted with 2 CV of 300 mM imidazole, 50 mM Tris-HCl pH 8.0, 300 mM NaCl. Imidazole was removed using PD-10 columns (GE Healthcare) equilibrated with 50 mM Tris-HCl, pH 8.0, 25 mM NaCl. The enzyme was stored at −20 °C. Thaw-freeze cycles that could alter the enzymes activity were not performed.

For larger-scale production of proteins (BnCE20_I, PpCE20II, SsCE20_IV, SpCE20_V, PwCE20_VI), 50 ng of each vector was transformed by heat shock into 50 μL of *E. coli* (strain BL21(DE3) pLysS for BnCE20_I and SpCE20_V; or *E. coli* SHuffle for PpCE20_II, SsCE20_IV and PwCE20_VI) competent bacteria. The plates were incubated for 12 h at 37 °C, and one of the colonies was transferred into a tube containing 5 mL liquid LB culture medium with 35 μg mL$^{-1}$ of kanamycin. The tubes were incubated under shaking at 200 rpm at 37 °C for 16 h (pre-inoculum).

For each protein, 2 L of liquid LB medium, distributed in 2 L Erlenmeyer flasks containing 500 mL of medium each, with 35 μg mL$^{-1}$ of kanamycin, were inoculated with 25 mL pre-inoculum and grown at 200 rpm and 37 °C until they reach an optical density of 0.6 at a wavelength of 600 nm (OD600), measured in a spectrophotometer (Spectronic Genesys 2). Induction was performed by addiction of 0.5 mmol L$^{-1}$ of isopropyl-β-D-thiogalactopyranoside (IPTG) (Promega) and the culture was incubated for 16 h at 20 °C under shaking at 200 rpm. The suspension was later centrifuged (Sorval, model RC5 Plus) for 10 min at 4 °C.

The pellet was resuspended in 100 mL of lysis buffer (20 mmol L$^{-1}$ phosphate, pH 7.4, 150 mmol L$^{-1}$ NaCl, 5 mmol L$^{-1}$ imidazole) and sonicated (Sonics) 10 times with 30 s pulses at 30-second intervals in an ice bath, with an amplitude of 30%. For preparation of the enzymes for crystallisation 1 mmol L$^{-1}$ PMSF was additionally added to the lysis buffer. No PMSF was added when the enzymes were used for activity assays. The lysate was centrifuged for 30 min at 10,000 x *g* and 4 °C, and the proteins were purified from the soluble extract by nickel affinity chromatography. Purification was carried out on FPLC equipment (Fast Protein Liquid Chromatography; GE Healthcare Life Sciences), using a 5 mL HiTrap Chelating column (GE Healthcare Life Sciences). Lysis buffer was used as Buffer A and the same solution, with the addition of 1 mol L$^{-1}$ imidazole, as Buffer B. Purification was carried out under continuous flow of 1.0 mL min$^{-1}$ (wash step: 10 column volumes, buffer A; wash: 2 column volumes, 5% buffer B; gradient elution: 20 column volumes, 5 to 50% buffer B). Sample collection was monitored at UV absorption at 280 nm.

After 13% SDS-PAGE analysis, samples were purified by size-exclusion chromatography by using a Superdex 200 16/60 column (GE Healthcare Life Sciences) previously equilibrated with 20 mmol L$^{-1}$ Hepes buffer pH 7.5, 150 mmol L$^{-1}$ NaCl, using FPLC equipment, and an isocratic flow of 0.5 mL min$^{-1}$. The eluted samples were evaluated in 13% SDS-PAGE, and those with the highest amount of protein and the highest degree of purity were used for biochemical experiments and crystallisation trials.

### Real-time structural motif search

Triads structurally similar to the water-mediated triad were searched with the advanced search feature "structure motif" at the RCSB[49]. Search motif was the Ser-His-Asp triad of Fl8CE20_II with Asn, Glu and Gln exchanges allowed for Asp and the default RMSD cutoff of 2 Å. 513 assemblies were found of which all had an Asp in the third position. After removal of all triads with a direct hydrogen bond His-Asp (shorter than 3 Å), structures of 18 different proteins remained. These were checked, if the Ser was a potential nucleophile, if the Ser hydrogen bonded to the NE2 of the His and if a water could bridge the His-Asp gap by hydrogen bonds. The requirement for the SerOG-HisNE2

hydrogen bond, rather than a SerOG-HisND1 hydrogen bond was based on the Nepsilon rule[49]. Only the sialic acid-*O*-acetyltransferase (XacXaeA) was identified by these criteria.

## Sequence similarity network

The Enzyme Function Initiative-Enzyme Similarity Tool was used to construct the SSN with the sequences from CE20 family. XacXaeA was used as a seed to find homologous amino acid sequences from Uni-Prot. Retrieved sequences were aligned by MUSCLE and used to build the hidden Markov Model (HMM) by HMMER. To reduce redundancy, sequences were grouped with limits from 45 to 90% identity using CD-Hit, resulting in 690 sequences. The initial SSN was made so it would connect sequences with an alignment score of at least 180. A cluster was categorised when containing at least five different sequences from five different organisms. Each sequence should be connected by at least 10 edges. The generated SSN was analysed by the Cytoscape software. Sequence domains were analysed by the Pfam webserver.

## Bioinformatic analyses of Fl8CE20_II

Genomes from Earth's Microbiomes (GEM) catalogue were reannotated using prokka (v1.14.6) and screened for Fl8 homologues using hmmsearch (v3.3.2, option settings: -E 1 --domE 1 --incE 0.01 --incdomE 0.03) with PFAM models PF03629.21 ("SASA") and PF02837.21 (Glyco_hydro_2_N). Sequences containing both domains were validated with hmmscan against the full Pfam-A database (as of 2024-03-18) using the profile's GA gathering cutoffs. Genomes encoding Fl8 homologues were then dereplicated using dRep (v3.4.2, option settings -l 0 -comp 50 -con 10 -pa 0.95 --checkM_method taxonomy_wf).

CAZymes encoding sequences ten genes up- and downstream of each Fl8 homologue were predicted using hmmscan vs. dbCAN-HMMdb-V12 and dbCAN-sub as well as diamond blastp (v2.1.1.155, option settings --evalue 1E-20 --id 30 --query-cover 40) against CAZyDB.07262023, provided by dbCAN. Results were further parsed using the hmmscan-parser script with an e-value cutoff of 1E-15 and a minimum coverage of 0.35. CAZyme families predicted by at least two tools were kept for further analysis.

Fl8 homologues were aligned using Clustal Omega (v1.2.4). Maximum-likelihood phylogeny was estimated with PhyML 3.0 webserver (default settings), and the resulting tree was visualised using iTOL.

## Protein crystallography

Fl8CE20_II was expressed with an N-terminal 6xHis tag as previously described[28]. Cells were resuspended in buffer A (50 mM Tris, pH 8.0, 100 mM NaCl, 10 mM imidazole) and disrupted by ultrasonication (Bandelin electronic™ Sonopuls™, HD 2070 Homogenisator, KN 76 sonotrode, 50% power, 0.5 pulse). After pelleting the cell debris by centrifugation (10,000 x *g*, 30 min, 4 °C), the supernatant was applied to a Nickel-sepharose column (Cube Biotech) equilibrated in the lysis buffer. Washing the column was also performed with lysis buffer. The enzyme was eluted in a gradient with buffer B (50 mM Tris, pH 8.0, 100 mM NaCl, 500 mM imidazole). Pooled protein fractions were concentrated to 3 ml and applied on a gel filtration column (Superdex200, 16/60, GE Healthcare) equilibrated and run with buffer A. Crystals were obtained from sitting drops of 0.3 µL protein solution (9.3 mg/mL) and 0.3 µL well solution (0.2 M Ca-acetate, 0.1 M Na-cacodylate, 18% PEG8000, Hampton Screen I, condition 46). After one week, crystals were harvested by a quick soak in 0.2 M Ca-acetate, 0.1 M Na-cacodylate, 22% PEG8000, 8% PEG400 and cryocooling in liquid nitrogen. Diffraction data was collected on beamline P13 at DESY, Hamburg, with a single wavelength experiment for a full 360° rotation in fine slices (0.1° oszillation images). The structure was solved by molecular replacement using an AlphaFold2 model for Fl8 with the structure split into the catalytic and the ancillary domain. Modelling and refinement were performed with coot and refmac5 in ccp4. As a

modification of the active site Ser (S112), acetylation, phosphorylation and cacodylation were tested. Later was confirmed by the high anomalous signal at the As position. Refinement included TLS, no NCS and occupancy refinement for 35 residues. Refinement statistics are also given in Supplementary Table S3.

For PpCE20_II samples were concentrated after size exclusion chromatography (100 mg mL⁻¹) and incubated for 1 hour with five times its molar concentration of PMSF, aiming for covalent inhibition. Samples were screened throughout initial conditions by the sitting-drop vapour diffusion technique, in 96-well plates. Drops were mixed in equal proportions of crystallisation solution: sample (1:1). Each drop was dispensed automatically by the HoneyBee 963 robot (Genomic Solutions), into 96-well plates, and then equilibrated at 18 °C, over a reservoir containing 80 µL. In total, 544 initial conditions were tested from commercial crystallisation kits: Crystal Screen and Crystal Screen 2 (Hampton Research), Wizard Screens I and II (Emerald), PACT, JCSG, SaltRx (Hampton Research), and Precipitant Synergy.

Diffraction data were collected at the MANACÁ beamline at the National Synchrotron Light Laboratory (LNLS-Sirius), Campinas-SP. The crystals were flash-frozen in a stream of nitrogen at 100 K prior to collection. X-ray diffraction data were acquired by the Pilatus 2 M detector (Dectris) mounted on a marDTB table, with an exposure time of 0.1 s (MANACÁ) and angular oscillations of 0.1° per image. At least a total of 3600 images were acquired for each crystal.

The data was indexed, integrated and scaled using the XDS software package. The collected data was analysed by the Xtriage programme to detect possible crystallographic pathologies. Molecular replacement was performed using the Phaser programme and the atomic coordinates of the crystallographic model of the homologous protein (XacXaeA PDB 7KMM). The refinement cycles were carried out using the programmes AutoBuild and PHENIX refine for automatic refinement, and COOT, to adjust the models manually based on the electron density maps. Local mobility was evaluated by the TLSMD programme and the information was introduced into the refinement cycles. Model quality was assessed by Molprobity. The crystallographic model figures were generated with the PyMOL Molecular Graphics System, version 1.8.0.2, Schrödinger, LLC.

## *p*-Nitrophenyl-acetate (pNPA) assay of Fl8CE20_II

The esterase activity was determined using final concentrations of 1 mM *p*NP-acetate, 50 mM buffer (standard Tris/HCl pH 8.0 if not stated otherwise), 100 mM NaCl, 5% DMSO and 25 µg/mL of the respective enzyme. The product formation was monitored continuously for 15 min every 30 sec, measuring the absorption at 348 nm[50] at RT. The slope in the linear area of the assay was used for calculation of specific activities ($\varepsilon = 15\,000\,M^{-1}\,cm^{-1}$). pH profiles were determined using 50 mM of the respected buffers instead of Tris/HCl. NaCl and metal ion profile was determined using variations of the reaction buffer with respective amounts of the additive. The kinetic assay was performed using final concentrations of 0.05, 0.1, 0.2, 0.5, 1, 2, 4 mM *p*NP-acetate and 50 mM Tris-HCl, pH 8.0, 100 mM NaCl and 5% DMSO at RT. The enzyme concentration was set to 3 µg/mL after determination of protein concentration using Pierce™ BCA Protein Assay Kit (Thermo Scientific™). The calculation of kinetic parameters was performed using the non-linear curve fit function (Michaelis-Menten) of OriginPro®. All experiments were performed in triplicate.

## *p*-Nitrophenyl-acetate (pNPA) assay of other CE20 enzymes

The hydrolysis of synthetic substrates was carried out using the batch method of formation of *p*-nitrophenolate from *p*-nitrophenyl acetate (pNP). Preferential temperature data was obtained using pNP-acetate in a range from 0 °C to 90 °C (or until activity was zero). Preferential pH data were obtained in the pH range of 2.5 to 11 using buffer suitable for each pH range, in a final concentration of 150 mmol L⁻¹ (2.5, 3.0 and 3.5: Gly/HCl; 3.5, 4.0, 4.5, 5.0 and 5.5: acetate/acetic acid; 5.5, 6.0, 6.5,

7.0, 7.5 and 8.0: phosphate; 8.5, 9.0, 9.5, 10.0, 10.5: Gly/NaOH; 10.5 and 11.0: carbonate/bicarbonate). Kinetic parameters reactions were carried out at the preferred temperature and pH of each enzyme, varying the concentration of the pNP-acetate substrate, in a final volume of 100 μL, under agitation at 400 rpm. The total amount of enzyme and the length of the reaction were estimated according to their linearity in protein concentration time. The amount of product formed after stopping the reaction was calculated from a standard curve of *p*-nitrophenolate (Sigma) under conditions similar to those used for each enzyme. The assays were performed in at least triplicates of each biological duplicate. The reaction was stopped by adding 100 μL acetonitrile (Sigma). After homogenisation, each reading was performed using 100 μL of the mixture and the product formed was quantified at 348 nm using a spectrophotometer (TECAN). Controls without enzyme were carried out to evaluate the spontaneous hydrolysis of substrates under the test conditions.

### Investigation of xylanolytic activities of Fl8CE20_II

Xylanolytic activities were determined as described previously[28]. Acetylated xylan from birchwood (AcX, Megazyme Ltd.) was pre-digested with a previously purified GH10 β−1,4-xylanase (UniProt A0A1D7XPY9) overnight. The enzyme and large polysaccharides were removed using a 10 kDa cutoff filter to obtain the pure acetylated oligosaccharides (AcX predigest). The enzyme (30 μg mL$^{-1}$) was incubated with the substrates at room temperature for 16 h using 50 mM Tris-HCl, pH 8.0, 100 mM NaCl. The detection of released acetic acid after the reaction was carried out using the Enzytec™ *Liquid* Acetic acid Kit from R-Biopharm (Darmstadt, Germany). The protocol was downscaled to the microtiter plate format by using 1/10$^{th}$ of the recommended volumes. As a control, the substrates were incubated without the enzyme. The obtained values were subtracted before calculation of the acetic acid concentration.

The investigation of esterase activity for glucuronoyl esterase was conducted using the K-URONIC assay from Megazyme and methyl-D-glucuronic acid (Biosynth) as a substrate[51]. The reaction was prepared using 100 μM enzyme and 40 mM methyl-D-glucuronic acid in 100 mM sodium phosphate buffer, pH 6, as recommended. The reaction was stopped after 30 mins by changing the temperature to 4 °C. Detection solution (50 μL) was added to the reaction (200 μL) in a microtiter plate and absorbance at 340 nm was measured for 60 min every 30 sec. A standard of glucuronic acid was included upon measurement to determine the time when the detection reaction finished. The change in absorption after subtracting the values of the negative control, incubated without enzyme, was determined. No change was observed for Fl8CE20_II.

Potential feruloyl esterase activity of the enzymes was assessed using ferulic acid methyl ester as a substrate. The enzyme (100 μg mL$^{-1}$) was incubated with ferulic acid methyl ester (0.4 mM) in 100 mM Tris-HCl, pH 8.0, 100 mM NaCl for 16 h at room temperature upon shaking at 1000 rpm. The reactions were stopped by heating to 80 °C for 10 min. For extraction, 50 μL of methanol was added to each reaction, vortexed for 30 sec, and then centrifuged at 13,000 x *g* for 5 min. The measurement was performed by injecting 5 μL into the uHPLC (column: Kinetex 2.6 μm, C18, 100 Å) using 80% water and 20% acetonitrile with 0.1% formic acid as the mobile phase. Absorption detection was carried out at 320 nm. Ferulic acid formation was calculated with a standard curve. No formation was detected for Fl8CE20_II. All experiments were performed in technical triplicates. Mean values and standard deviations were calculated.

### Mass spectrometry

To confirm that the abnormal density observed at position 112 (catalytic Ser 112) of Fl8CE20_II is an artefact of the crystallisation due to preparation upon usage of cacodylate buffer, the purified but not crystallised protein was analysed using mass spectrometry. As the densities of cacodylation are similar to phosphorylation, the protein was analysed in order to exclude a potential natural phosphorylation as a reason for the density observed in the crystal structure. The protein was digested in solution using trypsin and AspN as follows: 10 μg protein sample were reduced with 5 mM tris(2-chlorethyl)phosphate for 45 min at 60 °C and subsequently alkylated with 10 mM iodoacetamide for 15 min at ambient temperatures in the dark. Trypsin was added in a 1:100 enzyme: protein ratio, and the samples were incubated at 37 °C for 5 h. After addition of ZnCl$_2$ to yield a final concentration of 2.5 mM and AspN in a 1:100 enzyme: protein ratio, digestion was performed overnight at 27 °C. The reaction was stopped by adding 10 μL concentrated HCl and incubation at 4 °C for 30 min. After centrifugation at 10.000 x *g* at 4 °C for 10 min, the sample was subjected to ZipTip purification (Pierce C18 tips, 100 μL, Thermo Fisher Scientific) according to the manufacturer with the only exception that no further solvent was added at the beginning, as the sample has already been acidified before. Peptides were then loaded on an EASY-nLC 1200 system (Thermo Fisher Scientific) equipped with an in-house built 20 cm column (inner diameter 100 μm, outer diameter 360 μm) filled with ReproSil-Pur 120 C18 reversed-phase material (3 μm particles, Dr. Maisch GmbH). The column was inserted in an oven, securing a constant temperature of 45 °C. Peptides were eluted with a nonlinear 86 min gradient from 1 to 99% (v/v) solvent B (0.1% (v/v) acetic acid in 95% (v/v) acetonitrile) with a flow rate of 300 nL/min and injected online into an LTQ Orbitrap Elite (Thermo Fisher Scientific). The survey scan at a resolution of $R = 60,000$ and $1 \times 10^6$ automatic gain control target in the Orbitrap with activated lock mass correction was followed by selection of the 20 most abundant precursor ions for fragmentation. Singly charged ions as well as ions without detected charge states were excluded from MS/MS analyses.

For identification of peptides from MS-spectra, a database search was performed with MaxQuant 2.4.13.0 against an *E. coli* BL21 database obtained from Uniprot (UP000002032, downloaded 06/15/2021), which also contained the sequences of FL8 (total entries: 4156). Common laboratory contaminants and reverse entries were added by MaxQuant. MaxQuant was used with the following parameters: primary digest reagents, trypsin and AspN, N-terminal acetylation (+ 42.0106), oxidation on M (+ 15.9949), and phosphorylation on S or T (+ 79.9663) as variable modifications. Results were filtered for a 1% false discovery rate (FDR) on spectrum, peptide, and protein levels. Match between runs with default parameters was enabled. Results were filtered for 1% FDR on spectrum, peptide, and protein levels.

Targeted MS was performed on a Q Exactive instrument coupled to an EASY-nLC 1000 system (Thermo Fisher Scientific) equipped with an in-house built 20 cm column (inner diameter 100 μm, outer diameter 360 μm) filled with ReproSil-Pur 120 C18 reversed-phase material (1.9 μm particles, Dr. Maisch GmbH). Peptides were loaded with 0.1% (v/v) acetic acid at 500 bar and subsequently eluted with a 90 min gradient from 1 to 99% buffer B (0.1% (v/v) acetic acid in acetonitrile). The instrument was operated in selected ion monitoring (SIM) mode, where one survey scan (300 − 1650 Th mass range; 70,000 resolution at m/z 200; $3 \times 10^6$ predictive automatic gain control target; max. 200 ms injection time; activated lock mass correction) was followed by up to 10 fragment-scans (HCD at normalised energy for predefined precursor ions, mass range dependent on precursor m/z; 70,000 resolution at m/z 200; $1 \times 10^5$ predictive automatic gain control target; max. 200 ms injection time, isolation window 1.2 m/z, isolation offset 0.2 m/z).

All raw files from targeted MS were inspected in Skyline (version 23.1.0.380)[52]. In accordance with previously published criteria[53] peptides with a dot product of at least 0.7, a coeluting monoisotopic precursor ion, and at least 5 coeluting fragment ions (including a minimum of 3 fragment ions carrying the phosphorylation side) were considered as verified.

## Reporting summary

Further information on research design is available in the Nature Portfolio Reporting Summary linked to this article.

## Data availability

Atomic coordinates and structure factors have been deposited in the Protein Data Bank (PDB) with accession codes 9H4U (Deacetylase Fl8CE20_II from *Flavimarina* sp. Hel_I_48) and 9EGA (Deacetylase PpCE20_II from *Pedobacter psychrotolerans*). The structures of 8F9O (Dog sialic acid esterase (SIAE)), 1PPN (Monoclinic papain), 5LUB (Human legumain (AEP) in complex with compound 11), 2IRV (Rhomboid intramembrane serine protease GlpG), 6Y2E (SARS-CoV-2 main protease), 8B32 (Chalcone synthase from *Hordeum vulgare* complexed with CoA), 8IZJ (*E. coli* adenine phosphoribosyltransferase (APRT) in complex with AMP), 5MAL (Extracellular lipase from *Streptomyces rimosus*), 1ESC (*Streptomyces scabies* esterase), 8JUO (Aspartate semialdehyde dehydrogenase from *Porphyromonas gingivalis*) and 9D2D (*E. coli* cysteine desulfurase SufS R359A) were obtained from the PDB database. The processed data generated in this study are provided in the Source Data file. The raw data are available from the corresponding authors on request. Source data are provided in this paper.

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

## Acknowledgements

We thank the Brazilian Synchrotron Light Laboratory (LNLS – CNPEM/MCTI) for the provision of time at the MANACA beamline, the Brazilian Biosciences National Laboratory (LNBio – CNPEM/MCTI) for access to the crystallisation facility (Robolab), and the Brazilian Biorenewables National Laboratory (LNBR – CNPEM-MCTI) for the use of the Biophysics of Macromolecules facility. We thank the Deutsche For-schungsgemeinschaft (DFG) for funding this study in the frame of the research unit FOR 2406 (POMPU) through grants received by UTB (BO 1862/17-3), TS (SCHW 595/10-3 and 595/11-3) and DBe (BE 3869/4-3) and The São Paulo Foundation for Research (FAPESP) for funding through grants received by M.T.M. (21/04891-3) and post-doctoral fellowship to P.S.V. (2016/06509-0).

## Author contributions

M.T., P.S.V., M.T.M., M.L. and U.T.B. designed the study, supported by T.S. M.T. and P.S.V. performed the protein production, mutational studies and activity assays supported by T.Dö and T.Du. Protein crystallography was performed by G.J.P., P.S.V., supported by L.B. S.S.N. Analyses were carried out by G.F.P., while co-occurrence analyses and enzyme annotations were done by D.Ba. M.S. analyses were executed by SM and DBe. All authors reviewed and approved the manuscript.

## Funding

## Competing interests

The authors declare no competing interests.
