## [Transparent Peer Review file · Nature Communications]

Insights into a water-mediated catalytic triad architecture in CE20 carbohydrate esterases

Corresponding Author: Professor Uwe Bornscheuer

Version 0:

Reviewer comments:

Reviewer #1

(Remarks to the Author)

Insights into an undescribed catalytic triad architecture in CE20 carbohydrate esterase catalysis

the authors describe two carbohydrate esterase structures in an interesting manuscript. I like most of it but I am a bit concerned about what they called the "diluted catalytic triad" and I suggest to find a better name for it. Could it really be considered a triad?

I WOULD CHANGE THE TITLE INTO : Insights into an undescribed catalytic site architecture in CE20 1 carbohydrate esterase catalysis

Figure 2 has a black background possibly due to a conversion error please check and correct to a lighter background maybe white!

12 CE20 members operate via a diluted catalytic triad

WHAT IS THE MEANING OF DILUTED? PERHAPS WRITE IT AS CE20 MEMBERS OPERATE VIA A "DILUTED" CATALYTIC TRIAD aspartate, we termed this a diluted catalytic triad MAYBE BETTER CALL IT "WATER MEDIATED CATALYTIC TRIAD"? OR MAYBE IT CANNOT BE CALLED A CATALYTIC TRIAD? THE WATER IS PART OF THE CATALYTIC TRIAD AND NOT THE RESIDUE THAT COORDINATES THE WATER AM I WRONG?

Another support for this model is the fact that some CE20 enzymes are having the aspartate naturally replaced by an asparagine

BETTER REWRITTEN AS: ANOTHER SUPPORT FOR THIS MODEL IS THE FACT THAT IN SOME CE20 ENZYMES THE ASPARTATE IS NATURALLY REPLACED BY AN ASPARAGINE ANYWAY I AM SORRY BUT I DO NOT LIKE THIS TERM "DILUTED CATALYTIC TRIAD" CAN YOU USE A DIFFERENT WAY TO CALL IT??? A DILUTION IS FOR SOLUTIONS. A DILUTED SOLUTION OF NaCl.....

"As in the dyad serine proteases or in many cysteine proteases, the third residue is not 323 essential for an efficient catalysis.^{21,22}

The fact that the asparagine can replace the aspartate in several CE20 enzymes supports the model that the third residue in the catalytic triad is important to assist in the orientation of the catalytic histidine but not to activate the histidine side chain. Instead, it is mechanistically achieved indirectly by a coordinated water molecule in these enzymes.

YES I AGREE, SO WHY DO YOU CALL IT A CATALYTIC TRIAD? IT LOOKS MORE LIKE THE DYAD

This residue may be capable to act efficiently as a base to deprotonate the serine, increasing its nucleophilicity without the need to be polarized by a third catalytic triad residue. The main task of the aspartate/asparagine is therefore to orient the water molecule rather than to polarize and activate it for efficient catalysis.

AGAIN WHY CALL IT A CATALYTIC TRIAD WHEN TWO ARE CATALYTIC AND ONE IS HOLDING THE WATER IN PLACE?

“Thus, it is postulated”

POSTULATED IS FOR SOMETHING THAT CANNOT BE DEMONSTRATED RIGHT? MAYBE BETTER SAY:
PROPOSED?

This assumption is because the diluted catalytic triad is functional with an aspartate or asparagine binding and coordinating the water.

AGAIN CALL IT DIFFERENTLY IT IS NOT DILUTED!!! AND POSSIBLY NOT EVEN A TRIAD. AND IT IS BETTER TO
PROPOSE SOMETHING RATHER THAN ASSUMING IT

oscillation images = OSCILLATION IMAGES

Reviewer #2

(Remarks to the Author)

The manuscript addresses a report on carbohydrate esterases from a CE family. Based on the Methods section, the authors appear to have conducted significant work; however, not all the collected information is systematically presented or adequately described in the Results/Discussion sections. Certain aspects of the workflow or methodology are confusing, raising questions about why specific methods were used and how they were conducted. The manuscript seems rushed, with noticeable inconsistencies in writing style and repetitions across multiple sections. This suggests that multiple authors contributed to the document, but limited effort was made to ensure clarity, coherence, and flow. I found it difficult to follow, which significantly slowed my reading. Repetitions and unclear explanations, including for key points, hinder comprehension. Ultimately, I managed to read the entire manuscript in one sitting but noticed numerous issues with the writing, though not necessarily with the experiments themselves. I cannot endorse the manuscript in its current form. Below, I provide selected major comments rather than an exhaustive list.

1. Title and Introduction:

The title mentions an "underdescribed catalytic triad," but after reading the Abstract, Introduction, and initial sections of the Results, I could not discern what was previously known, what is novel, or what "underdescribed" means in this context. Additionally, some references cited in the Introduction are unclear due to odd numbering in the reference list, making it difficult to determine which sources the authors are referring to.

2. CE Families and Catalytic Triads:

According to the CAZy database, there are 20 CE families, as correctly stated in the Introduction. While it is acceptable not to list all families, the manuscript fails to clearly explain how catalytic triads differ among families. On page 3, conserved residues SGNH and the fold are mentioned, but this information appears to be well-established. This raises the question of what exactly makes the catalytic triad "underdescribed." Furthermore, there are four CE20 enzymes with crystal structures, yet it remains unclear how your protein of interest differs from these.

3. Source of the Enzyme:

From the beginning of the manuscript until late on page 5, there is no indication of the organism from which the enzyme was derived. This information is also absent in the Methods section, compounding the confusion.

4. Introduction—Last Paragraph:

The final paragraph of the Introduction should be rewritten to clearly articulate the research gap and hypothesis.

5. Repetitions in Results:

The first three sections of the Results contain significant repetitions, with the main points only becoming apparent later. While many details are provided in the figures, the text lacks sufficient explanations to complement them. Furthermore, certain experiments mentioned in the Methods do not have corresponding results presented. The Results section also includes speculative statements that are not adequately supported, such as in lines 142-149, 247, 282, and others. For example, in line 282, you state that a 4 Å distance "is too far away." Could this observation be an artifact of crystallization or other factors? Protein structures are inherently flexible, and while your interpretation may be scientifically valid, to what extent are you confident in your assumptions?

6. Discussion:

After reading the Discussion, I am still unclear about how your protein differs from others. Additionally, the Discussion introduces new data or information that is not mentioned in the Results, which further complicates the narrative.

Reviewer #3

(Remarks to the Author)

Tune et al. reported crystal structures of two full-length CE20 esterases involved in xylan breakdown. They found that the ancillary domain contributes to active site formation and classified all CE20 family enzymes into six clusters based on SSN. They also observed a water molecule mediating the interaction between H515 and D513 in the catalytic triad, and proposed

a “diluted catalytic triad” mechanism for catalysis with some mutagenesis data to support. Overall, the authors provided some new insights into the catalytic mechanism of CE20 enzymes and visualisation of the full length esterase structures could add to the understanding about the substrate binding. However, I have a few comments that need to be addressed.

Comments:

1. The authors classified the CE20 family into six clusters with potentially different substrate preferences. They speculated that the ancillary domain might be involved in substrate binding and specificity. Since the structures of CE20 (F18CE20_II and PpCE20_II) are resolved here, it would be beneficial to examine the similarity of residues in the active site/binding site among different clusters through sequence alignment.
2. The authors purified and crystallised two enzymes, F18CE20_II and PpCE20_II. However, they did not mention whether PpCE20_II is capable of processing xylan. Additionally, they did not perform activity assays for F18CE20_II on other carbohydrates (substrates for other clusters) to confirm that F18CE20_II and PpCE20_II are only xylan-processing enzymes. They also purified BnCE20_I, SsCE20_IV, and PwCE20_VI, and therefore could easily test their xylanolytic activity, as F18CE20_II.
3. A major concern is the use of PMSF during protein purification. PMSF is known to form adducts with catalytic serine residues, inhibiting enzymatic activity. The reported structure (9EGA) also shows PMSF-modified serine. Have the authors tested the activity of these enzymes in the absence of PMSF during purification?
4. The structure of 9H4U F18CE20_II shows a modified As-Ser112. The MS data confirming the modification (cacodylate or PMSF) is missing. An MS2 spectrum showing the modification is needed to validate this. Additionally, intact mass spectrometry would be a better tool to confirm the modification w/o cacodylate. The authors also used PMSF during the purification of F18CE20_II (Methods section was not detailed enough to tell whether PMSF was used during purification of F18CE20_II). Have they observed PMSF-modified S112 for F18CE20_II?
5. The PDB structure of 9H4U shows a water molecule in the density of As-Ser, which is likely incorrect.
6. The manuscript mentions that all three residues in the S-H-D triad are expected to be within 2–3 Å. What is the actual distance between S112 and H515 in this structure?
7. The distance between H515 and D513 is 4.0 Å at the resting state. Is it possible that during catalysis, histidine could directly contact between H515 and D513?
8. AlphaFold predictions for side-chain conformations are often less accurate. In this case, the results may not reliably indicate the presence of water in the active site.
9. In the mutagenesis study, the authors created D513A and D513N mutants. Both mutants could still accommodate a water molecule. Have the authors compared the activity of mutants with slightly bulkier residues, such as D513E or D513Q? This could provide insights because the water molecule would shift due to steric effects.
10. Fig3A, and 3G. Carbonyl oxygen of Q111 forms three hydrogen bonds in 3A and four hydrogen bonds in 3G, (particularly with another carbonyl oxygen of Q413)? This is not correct. Carbonyl oxygen can only form up to two hydrogen bonds. In addition, part of the residues are shown in sticks while some are shown in lines. Very confusing to see the interactions clearly.
11. The figures are bit of awkwardly organised, and sometimes it's difficult to follow. For example, the authors described the domain organisation of CE20 and classified CE20 according to SSN. However the SSN and classification is in Fig1 and domain architecture of CE20 only appears later in Fig.2a. The proposed mechanism appeared in Fig2d while the mutational evidence is in Fig3.
12. Fig3E, 3F. I believe Y334N (Fig3E) and Y334 (Fig3F) use carbonyl oxygen to interact with the water molecule. The carbonyl C and O are shown in sticks rather than lines as in Fig3C,D,H figures. In addition, labelling mutation as K313E in 3C, K313Q in 3D, Y334N in 3E, Y334Q in 3F, S434A in 3G and D513N in 3H make more sense than purple box.
13. Citation 17 is incorrectly numbered.

Version 1:

Reviewer comments:

Reviewer #1

(Remarks to the Author)

I thank the authors who provided a revised manuscript that took in consideration my suggestions and I hope it helped improving the manuscript also for other scientists interested in the topic

Reviewer #2

(Remarks to the Author)

-

Reviewer #3

(Remarks to the Author)

The authors have made substantial revisions to the manuscript and addressed most of my concerns.

However, my previous question regarding the xylanolytic activity of BnCE20_I, SsCE20_IV, and PwCE20_V remains unaddressed. The authors spent significant space (pages 6–10) on predicting substrate specificity based on sequence analysis and suggest that the ancillary domain of CE20 determines substrate preference. This is a key claim in the manuscript, particularly given that these structures differ from previously structures by having a fully resolved ancillary

domain (noted in Point 2 by Reviewer 2). However, this conclusion is largely speculative, as it lacks direct experimental evidence (no complex structures are provided, and in vitro assays are limited). Moreover, I do not see a clear connection between the sequence of the ancillary domain (including the loop region) and substrate selectivity based on the current data.

While I understand the challenges in synthesizing alternative substrates, it remains essential to test the xylanolytic activity of BnCE20_I, SsCE20_IV, and PwCE20_V using similar methods as those applied to F18CE20_II. Full kinetic characterization (e.g., k_{cat} and K_m) may not be necessary, but at least a qualitative assay should be conducted. The authors performed in vitro assays for F18CE20_II using ferulic acid and methyl-D-glucuronic acid (which showed no activity), yet they do not establish a clear link between this substrate selectivity and the ancillary domain.

Another major finding in the manuscript is the proposed water-mediated catalytic triad. With the new mutagenesis data combined with structural insights, this mechanism appears plausible.

Version 2:

Reviewer comments:

Reviewer #3

(Remarks to the Author)

The authors have addressed all my concerns. I would now support its publication in Nature Commu.

Point-by-point response to reviewer comments for manuscript NCOMMS-24-77350

First of all, we wish to thank the reviewers and to express our gratitude for the time and effort they have dedicated to evaluate our manuscript, especially over the Christmas and New Year holidays. We truly appreciate their commitment to support the academic community during this period. We believe that all of the suggestions made are really constructive and helped us to improve the manuscript. Please find below our detailed replies to the issues raised.

We very much hope that this detailed response, but especially the changes made to the manuscript (including additional experiments) are convincing so that an acceptance of our manuscript would be possible.

Reviewer 1:

Point 1:

The authors describe two carbohydrate esterase structures in an interesting manuscript. I like most of it but I am a bit concerned about what they called the "diluted catalytic triad" and I suggest to find a better name for it. Could it really be considered a triad?

We thank the reviewer for this comment. We have to stress that we extensively worked on the manuscript and we included some experimental data confirming the mechanism proposed. Furthermore, we provide additional sections in the introduction, results and discussion to clarify our decision of naming it a triad rather than a dyad. The denomination of this catalytic triad architecture as "diluted catalytic triad" was used to highlight that a water molecule, which is precisely coordinated, is necessary for efficient catalysis as it is putatively orienting and/or polarizing the catalytic histidine during catalysis. However, we agree the name might be misleading and therefore we changed it to "water-mediated catalytic triad". We also provide a detailed mechanistic explanation in the main manuscript. See also Point 6.

Point 2:

I WOULD CHANGE THE TITLE INTO: Insights into an undescribed catalytic site architecture in CE20 carbohydrate esterase catalysis.

We have changed the title to "**Insights into a water-mediated catalytic triad architecture in CE20 carbohydrate esterases**".

Point 3:

Figure 2 has a black background possibly due to a conversion error please check and correct to a lighter background maybe white!

We are sorry for this issue, this occurred during the upload process when the submission system generated the pdf file (it is not black when we made the pdf file from MS word). We have corrected it and checked this carefully when we submitted the revised manuscript.

Point 4:

CE20 members operate via a diluted catalytic triad WHAT IS THE MEANING OF DILUTED? PERHAPS WRITE IT AS CE20 MEMBERS OPERATE VIA A "DILUTED" CATALYTIC TRIAD. MAYBE BETTER CALL IT "WATER MEDIATED CATALYTIC TRIAD"? OR MAYBE IT CANNOT BE CALLED A CATALYTIC TRIAD? THE WATER IS PART OF THE CATALYTIC TRIAD AND NOT THE RESIDUE THAT COORDINATES THE WATER AM I WRONG?

We fully agree and call it "water-mediated catalytic triad".

Point 5:

Another support for this model is the fact that some CE20 enzymes are having the aspartate naturally replaced by an asparagine

BETTER REWRITTEN AS: ANOTHER SUPPORT FOR THIS MODEL IS THE FACT THAT IN SOME CE20 ENZYMES THE ASPARTATE IS NATURALLY REPLACED BY AN ASPARAGINE ANYWAY I AM SORRY BUT I DO NOT LIKE THIS TERM "DILUTED CATALYTIC TRIAD" CAN YOU USE A DIFFERENT WAY TO CALL IT??? A DILUTION IS FOR SOLUTIONS. A DILUTED SOLUTION OF NaCl.

We fully agree and call it "water-mediated catalytic triad". Regarding the designation as a triad rather than a dyad, please refer to the detailed mechanistic explanation in the main manuscript

Point 6:

“As in the dyad serine proteases or in many cysteine proteases, the third residue is not 323 essential for an efficient catalysis.^{21,22}. The fact that the asparagine can replace the aspartate in several CE20 enzymes supports the model that the third residue in the catalytic triad is important to assist in the orientation of the catalytic histidine but not to activate the histidine side chain. Instead, it is mechanistically achieved indirectly by a coordinated water molecule in these enzymes.

YES I AGREE, SO WHY DO YOU CALL IT A CATALYTIC TRIAD? IT LOOKS MORE LIKE THE DYAD

Our wording in this section was misleading, we apologize. We systematically checked enzymes using a catalytic dyad. Inspection of reported dyad enzymes with Ser as nucleophile and His as base shows that these enzymes are not active without any residue/molecular entity functionally replacing the third triad residue at least in orienting the base His. We have analyzed several enzymes with varying catalytic dyads (mammalian sialic acid esterase, an extracellular lipase from *Streptomyces rimosus*, papain, the protease papain) as detailed in the main manuscript (Supplementary Figures 13 and 14). Our mutational investigation clearly showed that the third triad residue, i.e. Asp515 in FI8CE20_II from *Flavimarina* sp., is important for catalysis. The mutation of Asp513 to Ala completely switches-off catalytic activity. As this might be due to its impact on substrate binding rather than catalysis, we created the mutant D513E (in FI8CE20_II), with Glu conserving the negative charge but being sterically more demanding compared to Asp. This mutant had only a mild impact on substrate binding lowering K_M by 3.7-fold (WT: 0.62 mM; D513E: 2.30 mM) while strongly impairing k_{cat} by 53-fold (WT: 9.70 s⁻¹; D513E: 0.18 s⁻¹). This shows that the third residue is, unlike in catalytic dyads, indeed important for catalysis and as it is too far to directly interact with the His Base. A direct contact of H515 and D513 is unlikely, also in course of the catalytic mechanism. Thus, our mutational data confirm the correct placement of the water molecule being indeed important for enzymatic activity.

Point 7:

This residue may be capable to act efficiently as a base to deprotonate the serine, increasing its nucleophilicity without the need to be polarized by a third catalytic triad residue. The main task of the aspartate/asparagine is therefore to orient the water molecule rather than to polarize and activate it for efficient catalysis.

AGAIN WHY CALL IT A CATALYTIC TRIAD WHEN TWO ARE CATALYTIC AND ONE IS HOLDING THE WATER IN PLACE?

Regarding the designation as a triad rather than a dyad, please refer to the response to point 6 and the revised sections in the main paper.

Point 8:

“Thus, it is postulated” POSTULATED IS FOR SOMETHING THAT CANNOT BE DEMONSTRATED RIGHT? MAYBE BETTER SAY: PROPOSED?

We fully agree and replaced “postulated” by “proposed”.

Point 9:

This assumption is because the diluted catalytic triad is functional with an aspartate or asparagine binding and coordinating the water.

AGAIN CALL IT DIFFERENTLY IT IS NOT DILUTED!!! AND POSSIBLY NOT EVEN A TRIAD. AND IT IS BETTER TO PROPOSE SOMETHING RATHER THAN ASSUMING IT

We agree with the reviewer and replaced the “diluted catalytic triad” by “water-mediated catalytic triad”. See our replies above.

Point 10:

oszillation images = OSCILLATION IMAGES

We corrected the word as suggested.

Reviewer 2:

Point 1:

Title and Introduction: The title mentions an "underdescribed catalytic triad," but after reading the Abstract, Introduction, and initial sections of the Results, I could not discern what was previously known, what is novel, or what "underdescribed" means in this context. Additionally, some references cited in the Introduction are unclear due to odd numbering in the reference list, making it difficult to determine which sources the authors are referring to.

Regarding your comment on the title, it seems there may have been a slight misunderstanding. In the title we had used the term "undescribed," but it appears it might have been read as "underdescribed." We hope this clarifies the intended wording. Please note that we have changed the title now to "**Insights into a water-mediated catalytic site architecture in CE20 carbohydrate esterases**". Nevertheless, we understand the concerns about not clarifying why this active site architecture is differing from known ones and have made substantial changes to the introduction and discussion sections (including further citations of mechanistically relevant publications) to provide more details on the current state of knowledge compared to our findings. Moreover, we addressed the issue of defining dyads and triad more precisely throughout the whole manuscript in order to provide a better understanding of our results.

Point 2:

CE Families and Catalytic Triads: According to the CAZy database, there are 20 CE families, as correctly stated in the Introduction. While it is acceptable not to list all families, the manuscript fails to clearly explain how catalytic triads differ among families. On page 3, conserved residues SGNH and the fold are mentioned, but this information appears to be well-established. This raises the question of what exactly makes the catalytic triad "underdescribed." Furthermore, there are four CE20 enzymes with crystal structures, yet it remains unclear how your protein of interest differs from these.

We have substantially revised the introduction to provide a much more detailed overview about the active sites of carbohydrate esterases representing different CE families. As mentioned in the comments to reviewer#1 and above, the introduction now clarifies the novelty in this active site architecture and its differences compared with other CE families in a better way.

"Furthermore, there are four CE20 enzymes with crystal structures, yet it remains unclear how your protein of interest differs from these."

The main difference between the existing structures and the two structures reported in our studies is that proteins in the CE20 family are not always having the ancillary domain as discovered in our work. In the Supplementary Figure 13 we compare one of the listed eukaryotic structures (8F9O), with the structure of F18_CE20. The structures of eukaryotic CE20 enzymes that can be found on CAZy do not have the ancillary domain (Ide et al., 2014). As we intended to clarify with this figure, the active site architecture of the CE20 enzymes without ancillary domains differs from the ones with ancillary domain (as in our enzymes). The structure of the CE20 family founding member XacXae (7KMM), which is part of the CE20 family with ancillary domain, could not be solved completely, as the ancillary domain was missing in the solved crystal structure. Along that line, also other structures of CE20 enzymes that were recently reported share a similar catalytic core domain consisting of a central SGNH hydrolase domain flanked by an N- and C-terminal β -sandwich domain but they lack the ancillary domain inserted into the SGNH hydrolase domain. Our data show the ancillary domain being important for both substrate binding and catalysis. Besides these structural data we describe the so far undescribed catalytic mechanism now called water-mediated catalytic triad. We refer to the main manuscript, where we provide more details to ensure a clearer line of argumentation and to make it more comprehensible for the reader.

Point 3:

Source of the Enzyme:

From the beginning of the manuscript until late on page 5, there is no indication of the organism from which the enzyme was derived. This information is also absent in the Methods section, compounding the confusion.

The reviewer is right. We added the information on the source of the enzymes structurally and functionally characterized in this manuscript.

Point 4:

Introduction—Last Paragraph:

The final paragraph of the Introduction should be rewritten to clearly articulate the research gap and hypothesis.

The entire introduction was rewritten, see revised manuscript.

Point 5:

Repetitions in Results:

The first three sections of the Results contain significant repetitions, with the main points only becoming apparent later. While many details are provided in the figures, the text lacks sufficient explanations to complement them. Furthermore, certain experiments mentioned in the Methods do not have corresponding results presented. The Results section also includes speculative statements that are not adequately supported, such as in lines 142-149, 247, 282, and others. For example, in line 282, you state that a 4 Å distance "is too far away." Could this observation be an artifact of crystallization or other factors? Protein structures are inherently flexible, and while your interpretation may be scientifically valid, to what extent are you confident in your assumptions?

We agree with the reviewer and reworked the whole results sections extensively in order to gain more clarity about the aim of the study and its main findings. We hope that this revision is now satisfactory.

"Protein structures are inherently flexible, and while your interpretation may be scientifically valid, to what extent are you confident in your assumptions?"

Our mutational data confirmed the correct placement of the water molecule being indeed important for enzymatic activity (Supplementary Table 5). We can exclude a direct interaction between the His base and Asp/Asn to occur during catalysis as a distance of $<3\text{Å}$ cannot be achieved rotating the side chains, i.e. χ_1 and/or χ_2 of the His base and/or the Asp/Asn. To this end, a conformational alteration of the main chain would be needed to allow a direct contact. However, this is unlikely to occur as the B-factors for the $C\alpha$ -atoms of the catalytic residues Ser-His-Asp are all between 17.7 to 29.7 Å² for both chains confirming these are structurally well defined with a low degree of flexibility (Supplementary Table 4). Along this line, we even observe this conformation in different catalytic states, i.e. in the apo-state, i.e. pre-catalysis state, represented by the unmodified chains and in the intermediate states, represented by the cacodylated/PMS-modified chains.

Point 6:

Discussion:

After reading the Discussion, I am still unclear about how your protein differs from others. Additionally, the Discussion introduces new data or information that is not mentioned in the Results, which further complicates the narrative.

We reworked the discussion section extensively to improve its readability and to enable a better understanding on what is new and how these structures differ from known CE20 structures and other carbohydrate esterases including citation of new references of related enzymes.

Reviewer 3:

Point 1:

The authors classified the CE20 family into six clusters with potentially different substrate preferences. They speculated that the ancillary domain might be involved in substrate binding and specificity. Since the structures of CE20 (F18CE20_II and PpC20_II) are resolved here, it would be beneficial to examine the similarity of residues in the active site/binding site among different clusters through sequence alignment.

We appreciate the helpful input, which has provided new insights and helped to strengthen our hypothesis. We performed sequence alignments of the selected CE20 sequences from the SSN to highlight the conservation of residues among the investigated CE20 enzymes. (Supplementary Figure 8) Additionally, we performed sequence alignments for each cluster to examine the conservation of residues within a loop region near the active site, which we propose is involved in substrate binding. (Supplementary Figure 9) We observed that, in the clusters (except for cluster I and IV), this loop region contains several, highly conserved amino acids which vary in each cluster. Due to the position of the loop, reaching into the active site, the cluster wise conserved residues presumably play a role in recognizing different substrates. We have added a figure (Supplementary Figure 9) showing the active site of a representative from each cluster, with the loop region and the corresponding conserved residues highlighted.

Point 2:

The authors purified and crystallised two enzymes, F18CE20_II and PpC20_II. However, they did not mention whether PpCE20_II is capable of processing xylan. Additionally, they did not perform activity assays for F18CE20_II on other carbohydrates (substrates for other clusters) to confirm that F18CE20_II and PpCE20_II are only xylan-processing enzymes. They also purified BnCE20_I, SsCE20_IV, and PwCE20_VI, and therefore could easily test their xylanolytic activity, as F18CE20_II.

We understand the concerns regarding experimental evidence for the hypothesis that the SSN clusters could be grouped according to their substrate specificity. Unfortunately, there are no commercially available acetylated substrates other than (synthetically) acetylated xylan. A detailed investigation of substrate specificity, which would require prior synthesis of model substrates, is beyond the scope of our current study as we are primarily focusing on the structural analysis of the ancillary domain containing CE20 enzymes, especially their active sites. Considering the low sequence similarities among clusters, especially with respect to the ancillary domain, such a study would be very time-consuming and beyond the scope of this manuscript. Our major findings are that we have discovered a new mechanism (the “water-mediated catalytic triad”) for F18CE20_II, determined the protein structures, and confirmed activity on xylan. This makes the current manuscript a full story. To get more insights into the substrate scope/specificity of F18CE20_II towards xylan, we added a proteome dataset from a previous study that shows upregulation of the F18CE20_II gene upon growth of the *Flavimarina* sp. Hel_I_48 strain on different xylans. As a negative control pectin, as another acetylated polysaccharide, was used and this showed no upregulation. It has to be mentioned that the protein could only be detected in one out of three proteome datasets generated upon growth on pectin (Supplementary Figure 4).

Point 3:

A major concern is the use of PMSF during protein purification. PMSF is known to form adducts with catalytic serine residues, inhibiting enzymatic activity. The reported structure (9EGA) also shows PMSF-modified serine. Have the authors tested the activity of these enzymes in the absence of PMSF during purification?

We are sorry for the misunderstanding due to improper differentiation in the method section. Only during the purification, i.e. during cell lysis, for crystallization of PpCE20_II PMSF was added. No PMSF was added when proteins were purified for enzyme assays. We rephrased the method section for clarification.

Point 4:

The structure of 9H4U F18CE20_II shows a modified As-Ser112. The MS data confirming the modification (cacodylate or PMSF) is missing. An MS2 spectrum showing the modification is needed to validate this. Additionally, intact mass spectrometry would be a better tool to confirm the modification w/o cacodylate. The authors also used PMSF during the purification of F18CE20_II (Methods section was not detailed enough to tell whether PMSF was used during purification of F18CE20_II). Have they observed PMSF-modified S112 for F18CE20_II?

In contrast to the experiments for PpCE20_II, we did not add PMSF in case of F18CE20_II (neither during cell lysis nor during further steps of purification or crystallization). The modification (cacodylation) visible in the crystal structure is due to the crystallization condition, which contained cacodylate as precipitant. As cacodylate modification is not as usual as PMSF, which is known to covalently bind to the serine of enzymes applying a catalytic triad with Ser/Cys as nucleophile and was added for that exact purpose, we only performed MS analyses of

F18CE20_II and not PpCE20_II. Due to the similarity of cacodylate to post translational modifications such as phosphorylation, we wanted to ensure that there was no unknown modification of this enzyme and that the observed modification was indeed due to the crystallization conditions. Since this concern did not apply to PMSF, no MS analyses were conducted. The MS results for the unmodified F18CE20_II has been uploaded to a data repository (see login details at the end of this document).

Point 5:

The PDB structure of 9H4U shows a water molecule in the density of As-Ser, which is likely incorrect.

The water molecule addressed by the reviewer near Ser112 is correctly placed. The water molecule is apparently placed to be located near the modified Ser112 side chain. However, the occupancies are 70% for the modified Ser (As-Ser112) without the water molecule and 30% for unmodified Ser112 plus water molecule. This is true for both chains.

Point 6:

The manuscript mentions that all three residues in the S-H-D triad are expected to be within 2–3 Å. What is the actual distance between S112 and H515 in this structure?

We are sorry for this mistake and added the information (3.1 Å) to the Figure.

Point 7:

The distance between H515 and D513 is 4.0 Å at the resting state. Is it possible that during catalysis, histidine could directly contact between H515 and D513?

Our mutational data confirmed the correct placement of the water molecule being indeed important for enzymatic activity (Supplementary Table 5). We can exclude a direct interaction between the His base and Asp/Asn to occur during catalysis as a distance of <3Å cannot be achieved rotating the side chains, i.e. χ_1 and/or χ_2 of the His base and/or the Asp/Asn. To this end, a conformational alteration of the main chain would be needed to allow a direct contact. However, this is unlikely to occur as the B-factors for the C α -atoms of the catalytic residues Ser-His-Asp are all between 17.7 to 29.7 Å² for both chains confirming these are structurally well defined with a low degree of flexibility (Supplementary Table 4). Along this line, we even observe this conformation in different catalytic states, i.e. in the apo-state, i.e. pre-catalysis state, represented by the unmodified chains and in the intermediate states, represented by the cacodylated/PMS-modified chains.

Point 8:

AlphaFold predictions for side-chain conformations are often less accurate. In this case, the results may not reliably indicate the presence of water in the active site.

AlphaFold structure predictions are very accurate in backbone geometry. Moreover, also side chain rotamer conformation are mostly correctly predicted if the side chains are present in secondary structure elements or conformationally restricted loop regions (Jumper et al., 2021). However, the reviewer is right that it is not possible so far to predict water molecules in protein structures by AlphaFold. To address this limitation of AlphaFold, we present two experimentally determined structures which clearly show the precise position of the water molecule in the active site with a well-defined tetrahedral coordination geometry. In each crystal structure, two monomers are present that show the exact same placement of the water molecule.

Point 9:

In the mutagenesis study, the authors created D513A and D513N mutants. Both mutants could still accommodate a water molecule. Have the authors compared the activity of mutants with slightly bulkier residues, such as D513E or D513Q? This could provide insights because the water molecule would shift due to steric effects.

The reviewer addresses an important point. As outlined in the revised manuscript, we therefore created the mutants F18CE20_II D513E and D513Q. Please see Figure 3i and 3j for location of the residues in the structure. The mutant F18CE20_II D513E, conserving the negative charge but Glu being sterically more demanding than Asp with consequences also on positioning the active site water molecule. We observed this mutant does result in a mild impact on substrate binding lowering K_M by 3.7-fold (WT: 0.62 mM; D513E: 2.30 mM) while strongly impairing k_{cat} by 53-fold (WT: 9.70 s⁻¹; D513E: 0.18 s⁻¹). This supports Asp513 is indeed involved in catalysis next to its role on substrate binding supporting the validity of the water-mediated Ser-His-(H₂O-Asp/Asn) catalytic triad architecture. For D513Q we observe that it impairs catalytic activity and substrate binding. This indicates that indeed the negative charge is important at this position.

Point 10:

Fig3A, and 3G. Carbonyl oxygen of Q111 forms three hydrogen bonds in 3A and four hydrogen bonds in 3G, (particularly with another carbonyl oxygen of Q413)? This is not correct. Carbonyl oxygen can only form up to two hydrogen bonds. In addition, part of the residues are shown in sticks while some are shown in lines. Very confusing to see the interactions clearly.

The reviewer is correct, we missed to remove some distance measurements. We also missed to show the main chain of a glycine residue forming one of the hydrogen bonds. We provide redrawn figures to correctly show the interactions. We represented the main chain as lines, while side chains are shown as sticks. We added this information to the Figure legend for clarity.

Point 11:

The figures are bit of awkwardly organised, and sometimes it's difficult to follow. For example, the authors described the domain organisation of CE20 and classified CE20 according to SSN. However the SSN and classification is in Fig1 and domain architecture of CE20 only appears later in Fig.2a. The proposed mechanism appeared in Fig2d while the mutational evidence is in Fig3.

We worked extensively on the manuscript and we reorganized the presentation of the figures to enable a better overview:

1. We added the domain architecture (former Fig. 2a) to Fig.1.
2. We replaced the general domain architecture in Fig.2a with the one for F18 in particular.

We decided against illustrating the mechanism in Fig. 3 because, following the addition of the new mutations D513E and D513Q, the figure would have become overly large and confusing. Additionally, we wished to present the active site (Fig. 2c) together with the mechanism and hope for your understanding.

Point 12:

Fig3E, 3F. I believe Y334N (Fig3E) and Y334 (Fig3F) use carbonyl oxygen to interact with the water molecule. The carbonyl C and O are shown in sticks rather than lines as in Fig3C,D,H figures. In addition, labelling mutation as K313E in 3C, K313Q in 3D, Y334N in 3E, Y334Q in 3F, S434A in 3G and D513N in 3H make more sense than purple box.

We agree and reworked Fig.3 after adding the additional mutations D513Q and D513E.

Point 13:

Citation 17 is incorrectly numbered

We checked the references and corrected it.

MS repository data access:**Reviewer access details**

Log in to the PRIDE website using the following details:

Project accession: PXD060139

Token: aJBUylz2eQVO

Alternatively, reviewer can access the dataset by logging in to the PRIDE website using the following account details:

Username: reviewer_pxd060139@ebi.ac.uk

Password: Gvj16ckeKIPx

New references:

1. Helbert, W. Marine Polysaccharide Sulfatases. *Frontiers in Marine Science* **4**, (2017).
2. Blair, D. E., Schüttelkopf, A. W., MacRae, J. I. & van Aalten, D. M. F. Structure and metal-dependent mechanism of peptidoglycan deacetylase, a streptococcal virulence factor. *Proceedings of the National Academy of Sciences* **102**, 15429–15434 (2005).

3. Taylor, E. J. *et al.* Structure and Activity of Two Metal Ion-dependent Acetylxylyan Esterases Involved in Plant Cell Wall Degradation Reveals a Close Similarity to Peptidoglycan Deacetylases*. *Journal of Biological Chemistry* **281**, 10968–10975 (2006).
4. Schwartz, L. A. *et al.* Carbohydrate Deacetylase Unique to Gut Microbe Bacteroides Reveals Atypical Structure. *Biochemistry* **64**, 180–191 (2024).
5. Montanier, C. *et al.* The Active Site of a Carbohydrate Esterase Displays Divergent Catalytic and Noncatalytic Binding Functions. *PLOS Biology* **7**, e1000071 (2009).
6. Rao, S. T. & Rossmann, M. G. Comparison of super-secondary structures in proteins. *Journal of Molecular Biology* **76**, 241–256 (1973).
7. Wetlaufer, D. B. Nucleation, rapid folding, and globular intrachain regions in proteins. *Proc Natl Acad Sci U S A* **70**, 697–701 (1973).
8. The UniProt Consortium. UniProt: the Universal Protein Knowledgebase in 2025. *Nucleic Acids Research* **53**, D609–D617 (2025).
9. Vernet, T. *et al.* Structural and Functional Roles of Asparagine 175 in the Cysteine Protease Papain (*). *Journal of Biological Chemistry* **270**, 16645–16652 (1995).
10. Till, M. *et al.* Structure and function of an acetyl xylan esterase (Est2A) from the rumen bacterium *Butyrivibrio proteoclasticus*. *Proteins: Structure, Function, and Bioinformatics* **81**, 911–917 (2013).
11. Zhang, L. *et al.* Crystal structure of SARS-CoV-2 main protease provides a basis for design of improved α -ketoamide inhibitors. *Science* **368**, 409–412 (2020).
12. Jin, Z. *et al.* Structure of Mpro from SARS-CoV-2 and discovery of its inhibitors. *Nature* **582**, 289–293 (2020).
13. Peng, B. *et al.* Engineering a Plant Polyketide Synthase for the Biosynthesis of Methylated Flavonoids. *J. Agric. Food Chem.* **72**, 529–539 (2024).
14. Choi, W., Wu, H., Yserentant, K., Huang, B. & Cheng, Y. Efficient tagging of endogenous proteins in human cell lines for structural studies by single-particle cryo-EM. *Proceedings of the National Academy of Sciences* **120**, e2302471120 (2023).
15. Dahal, P., Pathak, D. & Kwon, E. Crystal structure of *Bacillus subtilis* glyceraldehyde-3-phosphate dehydrogenase GapB. **11**, 59–65 (2023).

16. Hlaing, S. H. S. *et al.* Strategy to Select an Appropriate Cryoprotectant for an X-ray Study of *Escherichia coli* GAPDH Crystals. *Crystal Growth & Design* **23**, 7126–7133 (2023).
17. Hwang, J., Do, H., Shim, Y.-S. & Lee, J. H. Crystal Structure of Aspartate Semialdehyde Dehydrogenase from *Porphyromonas gingivalis*. *Crystals* **13**, 1274 (2023).
18. Gogar, R. K. *et al.* The structure of the SufS–SufE complex reveals interactions driving protected persulfide transfer in iron-sulfur cluster biogenesis. *Journal of Biological Chemistry* **300**, (2024).
19. LaLonde, J. M. *et al.* Use of Papain as a Model for the Structure-Based Design of Cathepsin K Inhibitors: Crystal Structures of Two Papain–Inhibitor Complexes Demonstrate Binding to S'-Subsites. *J. Med. Chem.* **41**, 4567–4576 (1998).
20. O'Farrell, P. A. & Joshua-Tor, L. Mutagenesis and crystallographic studies of the catalytic residues of the papain family protease bleomycin hydrolase: new insights into active-site structure. *Biochemical Journal* **401**, 421–428 (2006).
21. Jia, Z. *et al.* Crystal Structures of Recombinant Rat Cathepsin B and a Cathepsin B-Inhibitor Complex: IMPLICATIONS FOR STRUCTURE-BASED INHIBITOR DESIGN (*). *Journal of Biological Chemistry* **270**, 5527–5533 (1995).
22. Mirković, B. *et al.* Novel Mechanism of Cathepsin B Inhibition by Antibiotic Nitroxoline and Related Compounds. *ChemMedChem* **6**, 1351–1356 (2011).

Point-by-point response to reviewer comments for manuscript NCOMMS-24-77350

We would like to thank all three reviewers once again for taking the time to evaluate our substantially revised manuscript. We truly appreciate the positive feedback and addressed the remaining concerns to the best of our ability in order to further improve the manuscripts quality.

Reviewer #1:

I thank the authors who provided a revised manuscript that took in consideration my suggestions and I hope it helped improving the manuscript also for other scientists interested in the topic

We are very grateful that reviewer #1 was pleased with the previous revision.

Reviewer #2 (Remarks to the Author):

-

Obviously, reviewer #2 was also pleased with the previous revision.

Reviewer #3 (Remarks to the Author):

The authors have made substantial revisions to the manuscript and addressed most of my concerns. However, my previous question regarding the xylanolytic activity of BnCE20_I, SsCE20_IV, and PwCE20_V remains unaddressed. The authors spent significant space (pages 6–10) on predicting substrate specificity based on sequence analysis and suggest that the ancillary domain of CE20 determines substrate preference. This is a key claim in the manuscript, particularly given that these structures differ from previously structures by having a fully resolved ancillary domain (noted in Point 2 by Reviewer 2). However, this conclusion is largely speculative, as it lacks direct experimental evidence (no complex structures are provided, and *in vitro* assays are limited). Moreover, I do not see a clear connection between the sequence of the ancillary domain (including the loop region) and substrate selectivity based on the current data.

We understand the concerns addressed by the reviewer about our hypothesis regarding the substrate preference-based clustering of the CE20s. We wanted to point out that the ancillary domains lining the active site show the major differences in the sequences of enzymes in the different clusters. This is why we hypothesized that the real physiological substrates of the enzymes might be different for the different clusters. But there is no doubt that this indeed needs further analyses. So we hopefully adapted our wording in the newly revised manuscript clearly enough to address this lack in experimental evidence (besides mentioned *in silico* analyses).

As suggested, we now have performed the requested deacetylase activity assays of the above-mentioned enzymes towards acetylated xylan. Please note that we had to replace the acetate detection kit with another one, as the previously used kit is unfortunately no longer available from the manufacturer (see changes in the Methods section). We could observe that the enzymes BnCE20_II, SsCE20_IV and PwCE20_VI you have mentioned are indeed also active on the semisynthetic acetylated xylan. We therefore decided to significantly shorten the section on substrate preference, removed the section about the loop to clearly emphasize once again that this is a hypothesis based primarily on *in silico* analyses, which requires further experimental validation, not provided in this study due to a lack of suitable model substrates. As we did not wish to remove the section entirely from the manuscript, we hope that the substantial shortening and rephrasing—along with a clear reference to the lack of experimental evidence—will be satisfactory.

In this context, we would like to emphasize that the main point (and scientific relevance) of our manuscript is the structure elucidation and especially the newly described “water-mediated” mechanism of these enzymes.

While I understand the challenges in synthesizing alternative substrates, it remains essential to test the xylanolytic activity of BnCE20_I, SsCE20_IV, and PwCE20_V using similar methods as those applied to F18CE20_II. Full kinetic characterization (e.g., *k*_{cat} and *K*_m) may not be necessary, but at least a qualitative assay should be conducted. The authors performed *in vitro* assays for F18CE20_II using ferulic acid and methyl-D-glucuronic acid (which showed no activity), yet they do not establish a clear link between this substrate selectivity and the ancillary domain.

We apologize for the misleading phrasing. We did not wish to suggest that the ancillary domain is linked to the esterase function itself (deacetylase/acetyl xylan esterase, feruloyl esterase, glucuronylesterase). The xylanolytic activity assays using ferulic acid and methyl-D-glucuronic acid were performed to clarify the deacetylase/acetylxylan esterase function of the F18CE20_II and to exclude activity as a feruloyl esterase or glucuronylesterase which is also common for xylan targeting esterases. This finding has no linkage to the ancillary domain at all, which we now clarified in the text again. According to our understanding, the ancillary domain varies solely based on the specific acetylated polysaccharide involved (xylan, xyloglucan etc.). We do not believe that the ancillary domain alters the esterase function itself— all

CE20 enzymes function as deacetylases and have so far not shown any other carbohydrate esterase activities. Thank you for pointing out this potential misunderstanding. Regarding the xylanolytic activity of BnCE20_I, SsCE20_IV, and PwCE20_V please see our comment above.

Another major finding in the manuscript is the proposed water-mediated catalytic triad. With the new mutagenesis data combined with structural insights, this mechanism appears plausible.

We are very grateful about this comment and appreciate very much that reviewer #3 agrees that our mechanism appears plausible as this is the major outcome and discovery of our study.

In conclusion, we would like to once again thank Reviewer 3 for his/her expertise, which has significantly contributed to improving our manuscript. It was indeed correct to shift the focus away from the highly speculative substrate specificity and instead direct it towards the central aspect of the manuscript, the mechanism.